# Receptor for advanced glycation end-products (RAGE) mediates phagocytosis in nonprofessional phagocytes

Yan Yang[1], Guoyu Liu[1], Feng Li[1], Lucas B. Carey [2,3], Changjin Sun[1], Kaiping Ling[1], Hiroyuki Tachikawa[4,5], Morihisa Fujita [1,6], Xiao-Dong Gao [1✉] & Hideki Nakanishi [1✉]

In mammals, both professional phagocytes and nonprofessional phagocytes (NPPs) can perform phagocytosis. However, limited targets are phagocytosed by NPPs, and thus, the mechanism remains unclear. We find that spores of the yeast *Saccharomyces cerevisiae* are internalized efficiently by NPPs. Analyses of this phenomenon reveals that RNA fragments derived from cytosolic RNA species are attached to the spore wall, and these fragments serve as ligands to induce spore internalization. Furthermore, we show that a multiligand receptor, RAGE (receptor for advanced glycation end-products), mediates phagocytosis in NPPs. RAGE-mediated phagocytosis is not uniquely induced by spores but is an intrinsic mechanism by which NPPs internalize macromolecules containing RAGE ligands. In fact, artificial particles labeled with polynucleotides, HMGB1, or histone (but not bovine serum albumin) are internalized in NPPs. Our findings provide insight into the molecular basis of phagocytosis by NPPs, a process by which a variety of macromolecules are targeted for internalization.

[1] Key Laboratory of Carbohydrate Chemistry and Biotechnology, Ministry of Education, School of Biotechnology, Jiangnan University, Wuxi 214122, China. [2] Center for Quantitative Biology and Peking-Tsinghua Center for Life Sciences, Academy for Advanced Interdisciplinary Studies, Peking University, Beijing 100871, China. [3] Ginkgo Bioworks, Boston, MA 02210, USA. [4] Department of Applied Biological Chemistry, Graduate School of Agricultural and Life Sciences, University of Tokyo, Tokyo 113-8657, Japan. [5] Collaborative Research Institute for Innovative Microbiology, University of Tokyo, Tokyo 113-8657, Japan. [6] Institute for Glyco-core Research (iGCORE), Gifu University, Gifu 501-1193, Japan. ✉email: xdgao@jiangnan.edu.cn; hideki@jiangnan.edu.cn

Diverse eukaryotic cells can engulf large particles (≥0.5 μm in diameter) and internalize them via an endocytic process called phagocytosis[1]. In mammals, a class of cells have evolved to perform phagocytosis; these cells, including macrophages, neutrophils, and dendritic cells, are termed professional phagocytes[1]. Apart from professional phagocytes, however, phagocytosis occurs in many other cells, such as epithelial cells, fibroblasts, and tumors; these cells are termed nonprofessional phagocytes (NPPs)[2–5]. Compared to professional phagocytes, the range of macromolecules internalized by NPPs is limited[3]. Nevertheless, studies using macrophage-deficient mice have demonstrated the functional redundancy between NPPs and macrophages, at least for the removal of apoptotic cells[6,7]. However, very few studies have been conducted to explore the physiological roles of phagocytosis by NPPs partly because the molecular mechanism is poorly understood.

Cells can internalize large particles via phagocytosis or macropinocytosis[1]. Macropinocytosis is a process by which extracellular molecules are randomly internalized in cells. In contrast, phagocytosis is a receptor- and ligand-induced endocytic process[1]. Thus, in this process, particles decollated with specific ligands are targeted for internalization. Various phagocytic receptors, such as Fcγ receptors, integrins, and scavenger receptors, are found in professional phagocytes[8]. In general, phagocytic receptors are activated by binding to their specific ligands, which leads to the reorganization of the actin cytoskeleton to deform the plasma membrane. When the plasma membrane is extended around the target particles, phagocytic receptors sequentially bind to their ligand presented on target particles. Eventually, the target particles are engulfed by the plasma membrane and internalized in phagocytic cells[1]. Internalized particles are compartmentalized in a membrane-bound structure termed the phagosome, which matures into phagolysosomes by fusion with lysosomes[9].

One objective of our study is to use spores of the yeast *Saccharomyces cerevisiae* as microparticles[10]. Yeast spores are a dormant and stress-resistant cell form that is generated when diploid cells are incubated under starvation conditions[11]. Spore formation occurs inside of the mother cells, where four nuclei produced via meiosis are individually enclosed by the spore plasma membrane and spore wall. Through this process, the mother cells become asci, including four spores. Unlike vegetative cells, spores have dityrosine and chitosan layers on the exterior of their cell (spore) wall[12]. Chitosan is often found in fungal cell walls, but the dityrosine layer is a unique structure found in the *S. cerevisiae* spore wall[13,14]. The primary constituent of the dityrosine layer is LL-bisformyl dityrosine. While the detailed structure of this layer remains unclear, LL-bisformyl dityrosine molecules are presumably cross-linked to produce a macromolecule which is covalently attached to the chitosan layer[15] The dityrosine and chitosan layers make spores resistant to environmental stresses[11].

During the course of our study, we performed an experiment to incubate spores with cultured mammalian cells; in this experiment, we found that spores are phagocytosed in NPPs. Further analysis of this phenomenon revealed that internalization of spores by NPPs is mediated by receptors for advanced glycation end-products (RAGE). RAGE is a member of the immunoglobulin receptor superfamily and was originally identified as a receptor for advanced glycation end-products[16,17]. However, RAGE is now known to recognize multiple ligands, including polynucleotides, phosphatidylserine (PS), and high mobility group box 1 (HMGB1)[18–20]. Most RAGE ligands are known as damage-associated molecular patterns (DAMPs), and activation of the RAGE signaling pathway is known to induce inflammation[21,22]. RAGE can internalize its ligands via the endocytic pathway[18,23]. In addition, RAGE can recognize PS and is involved in the phagocytosis of apoptotic cells (efferocytosis) in macrophages[19]. However, its role in phagocytosis in NPPs is unknown. Spores are phagocytosed in NPPs because a RAGE ligand, RNA, is attached to the spore wall. Apart from spores, particles decorated with RAGE ligands are phagocytosed in NPPs. Our results demonstrate that various macromolecules can be internalized in NPPs via RAGE-mediated phagocytic processes.

## Results

**Spores are internalized in NPPs**. Yeast spores are produced in the ascus (Supplementary Fig. 1a). By rupturing the ascal membrane and ascal wall, spores can be released from the asci (Supplementary Fig. 1b). When spores were incubated with HEK293T cells, we found that spores, but not vegetative yeast cells, exhibited toxicity to HEK293T cells (Supplementary Fig. 1c). Unlike vegetative cells, spores have dityrosine and chitosan layers on the exterior of their cell (spore) wall[12] (Supplementary Fig. 1a). Since vegetative cells were harmless (Supplementary Fig. 1c), we speculated that the toxicity of spores is attributable to the unique structure of the spore wall. Consistent with this hypothesis, *dit1Δ* spores, which lack the dityrosine layer[24] (Supplementary Fig. 1a), did not exhibit toxicity (Supplementary Fig. 1c). When HEK293T cells incubated with spores were observed under the microscope, we realized that spores were internalized in the mammalian cells. We confirmed that spores were incorporated into lysosomes by staining with an acidotropic probe LysoTracker (Fig. 1a, b). Spores were not stained by LysoTracker in the condition used in our internalization assay (Supplementary Fig. 1d). Compared to wild-type spores, the internalization levels of vegetative cells and *dit1Δ* spores by HEK293T cells were lower (Fig. 1c, d). We found that a fraction of spores was swollen in HEK293T cells when they were incubated for 12 h (Supplementary Fig. 1e), suggesting that spores could germinate in the cultured cells. To assess whether the viability of spores is related to their toxicity, heat-killed spores were incubated with HEK293T cells. The levels of spore internalization by HEK293T cells were not altered by heat treatment (Supplementary Fig. 1f); however, heat-killed spores did not exhibit toxicity (Supplementary Fig. 1g). These results suggest that the cytotoxicity of spores is attributable to their internalization and germination. Since *S. cerevisiae* is not a pathogenic organism, the physiological significance of the spore's toxicity is unclear. Nevertheless, we are interested in the phenomenon that spores are internalization in cultured cells; thus, the basis of the internalization mechanism was further analyzed.

Spores can be internalized in various cell lines, including HEK293, HeLa, and human uroepithelial cells (Fig. 1c). Among the cell lines we tested, HEK293T cells internalized spores most efficiently (Fig. 1c); thus, this cell line was used for further studies. Given that spores are ~3 μm in diameter (Supplementary Fig. 1b), they are likely internalized via phagocytosis or macropinocytosis. Because spores are targeted for internalization as described below, we hypothesized that spores are internalized via phagocytosis. Since the cell lines used in the internalization assay were all derived from epithelial tissues, these results suggested that spores induce phagocytosis in NPPs. Similar to Fcγ receptor-mediated phagocytosis in macrophages, internalization of spores in HEK293T cells was inhibited by inhibitors of actin polymerization[25], spleen tyrosine kinase (SYK)[26], and phosphatidylinositol-3 kinase (PI3K)[27] (Fig. 1e). In addition to the microscopy-based phagocytosis assay, inhibitory effect of the inhibitors in spore internalization by HEK293T cells was verified with fluorescence-activated cell sorting (FACS)-based phagocytosis assay (Supplementary Fig. 2). Expression of SYK in HEK293T cells was confirmed by western blotting

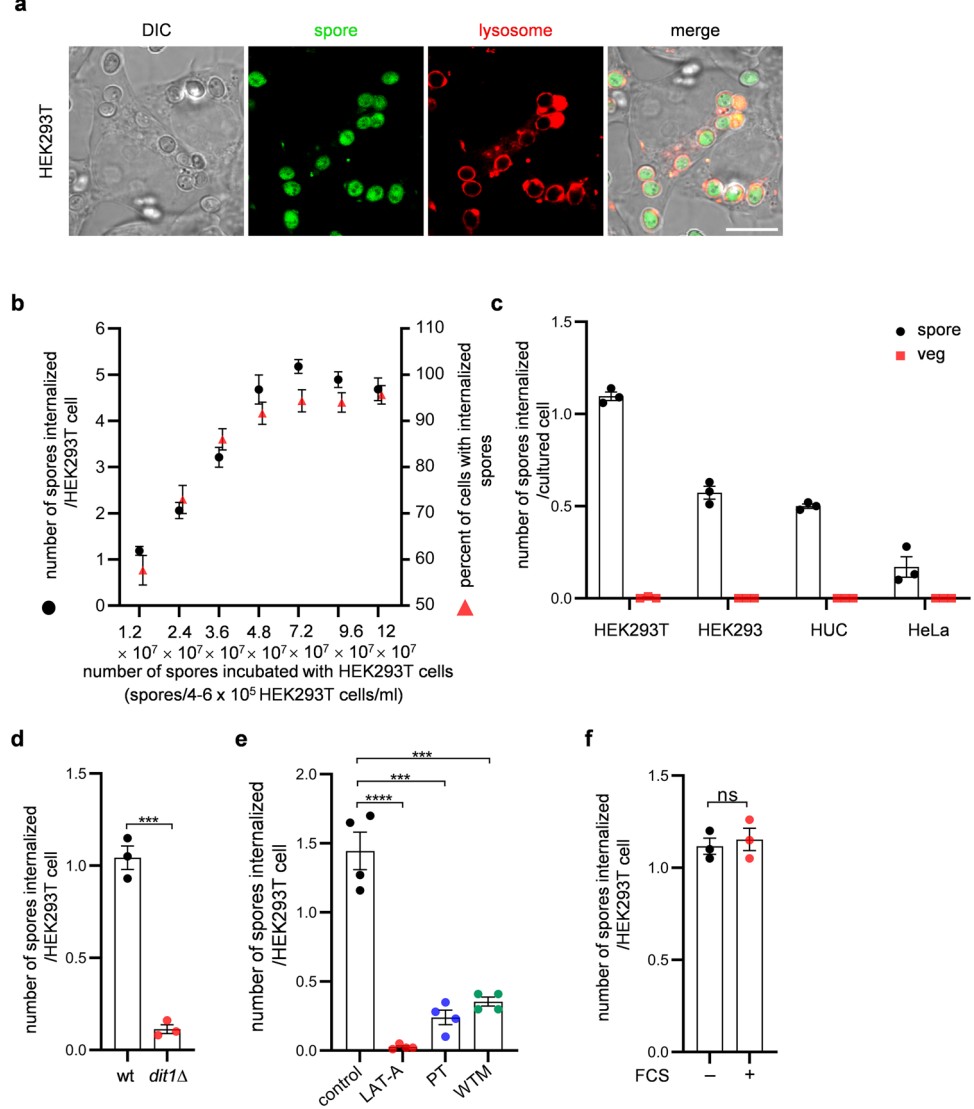

**Fig. 1 Spores are internalized by NPPs. a** Representative images of HEK293T cells with internalized spores expressing GFP was obtained with differential interference contrast (DIC) or fluorescence microscopy (spore and lysosome). HEK293T cells were incubated with spores at $3.6 \times 10^7$ spores/$4$–$6 \times 10^5$ HEK293T cells/ml. Lysosomes were stained with LysoTracker Red. Scale bar, 10 μm. **b** Numbers of spores internalized per HEK293T cell (black circles) and percentages of HEK293T cells with internalized spores (red triangles). HEK293T cells were incubated for 1 h with various quantities of spores in DMEM. **c** Internalization of yeast cells in NPPs. Spores or vegetative (veg) cells were incubated with HEK293T cells at $1.2 \times 10^7$ yeast cells/$4$–$6 \times 10^5$ HEK293T cells/ml; or with HEK293 cells, human uroepithelial cells (HUC), or HeLa cells at $1.2 \times 10^7$ yeast cells/$3.4$–$4.6 \times 10^5$ mammalian cells/ml. **d** Internalization of wild-type (wt) or *dit1Δ* spores in HEK293T cells. **e** Internalization of spores in HEK293T cells treated with or without (control) actin polymerization inhibitor latrunculin A (LAT-A), spleen tyrosine kinase inhibitor piceatannol (PT), or phosphatidylinositol-3 kinase inhibitor wortmannin (WTM). **f** Effect of the presence (+) or absence (−) of 10% fetal calf serum (FCS) on spore internalization in HEK293T cells. Data were presented as the mean ± SEM (**b**–**f**). Statistical significance was determined by two-tailed unpaired Student's *t*-tests. $n = 3$ (**b**–**d**, **f**), $n = 4$ (**e**). \*\*\*$P < 0.001$; \*\*\*\*$P < 0.0001$; ns, not significant ($P \geq 0.05$).

(Supplementary Fig. 3a). Furthermore, we found that the levels of spore internalization by HEK293T cells were decreased by *SYK* knockout (Supplementary Fig. 3b). Phagocytosis assays were performed in the media supplemented with heat-inactivated serum in this study. In addition, HEK293T cells internalized spores in the absence of serum (Fig. 1f). Thus, internalization of spores in HEK293T cells occurs even in the absence of antibodies and the complement system.

**RNA fragments are attached to the spore wall.** Spores are efficiently internalized in NPPs, suggesting that the spore wall is decorated with a ligand to induce phagocytosis. Given that the internalization of spores was inhibited by washing with a high-

salt (0.6 M NaCl) solution (Fig. 2a), the ligand is most likely noncovalently associated with the outermost layer (dityrosine layer) of the spore wall. Since spores are formed in the mother cell cytosol (Supplementary Fig. 1a), we speculated that cytosolic materials would attach to the spore wall. Strikingly, the levels of spore internalization by HEK293T cells were decreased by RNase treatment but not by DNase treatment (Fig. 2a). In line with the result, RNA was detected in the eluate of spores washed with high-salt solution (Fig. 2b). Internalization levels of spores were also decreased by protease treatment (Fig. 2a). Compared to the case with wild-type spores, washing of *dit1Δ* spores with high-salt solution released lower amounts of RNA from the spore wall (Fig. 2c), a result that correlated with the observation that

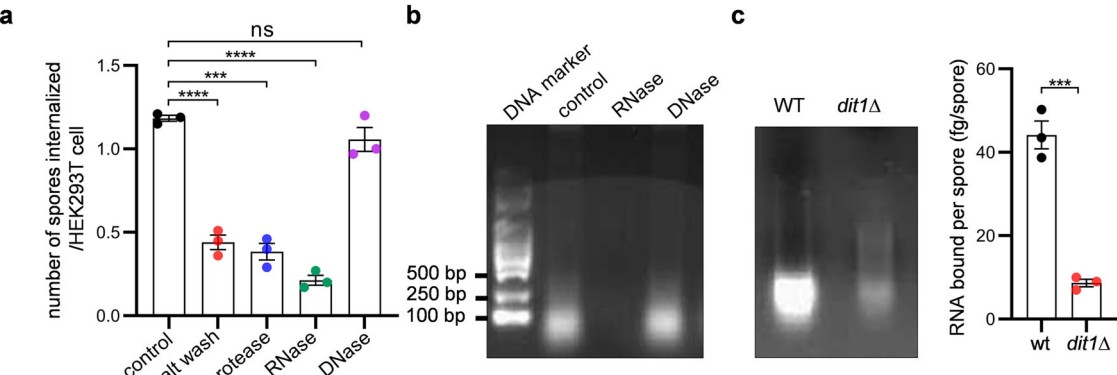

**Fig. 2 RNA is attached to the spore wall. a** Internalization of spores treated with indicated reagents in HEK293T cells. **b** Detection of RNA bound to the spore wall. Nucleic acids purified from high-salt eluate of spores were treated with indicated enzymes. These treated and untreated (control) samples were subjected to agarose gel electrophoresis and stained with a fluorescent nucleic acid dye. A double-stranded DNA marker was used as a reference. **c** Left panel: Detection of RNA bound to wild-type (wt) or $dit1\Delta$ spores. Right panel: Amount of RNA eluted from $2 \times 10^8$ of wild-type (wt) or $dit1\Delta$ spores. Data were presented as the mean ± SEM (**a**, **c**). Statistical significance was determined by two-tailed unpaired Student's $t$-tests. $n = 3$ (**a**, **c**). ***$P < 0.001$; ****$P < 0.0001$; ns not significant ($P \geq 0.05$).

internalization levels of spores by HEK293T cells were decreased by $dit1\Delta$ mutation (Fig. 1d). The FACS-based phagocytosis assay also showed that HEK293T cells containing spores were decreased when spores were treated with RNase, protease or high-salt solution (Supplementary Fig. 4).

Internalization of spores may be induced by a unique RNA attached to the spore wall. Thus, spore wall-bound RNA was identified by RNA sequencing. The majority of the RNA fragments released from the spore wall ranged from 25 to 29 nucleotides (nt) (Supplementary Fig. 5a). The RNA sequencing result (deposited in the Sequence Read Archive database, accession number PRJNA748575) showed that more than half of the RNA fragments were derived from tRNA sequences (Supplementary Fig. 5b). Notably, among tRNA-derived fragments, 87% were fragments derived from tRNA$^{Asp}$ species (Supplementary Fig. 5c). Apart from tRNA fragments, rRNA-, viral RNA-, and mRNA-derived fragments were found (Supplementary Fig. 5b). Thus, the spore wall is decorated with a variety of cytosolic RNA sequences.

**Polynucleotides serve as ligands to induce phagocytosis in NPPs.** Since tRNA was most abundantly found in spore wall-bound RNA, we analyzed whether phagocytosis by HEK293T cells is induced solely by tRNA. For this purpose, we assessed the internalization of tRNA-bound latex beads by HEK293T cells. Latex beads that contain surface amine groups were shown to adsorb purified tRNA (Fig. 3a). The levels of latex bead internalization by HEK293T cells were increased by the binding of tRNA (Fig. 3b). Latex beads were not stained by LysoTracker in the internalization assay (Supplementary Fig. 1d). However, the levels of internalization of polynucleotide-bound latex beads were lower than spores; as shown in Fig. 2a (control) and 3b (tRNA), when $1.2 \times 10^7$ beads (incubated with tRNA at $300\,\mu g/2 \times 10^8$ beads/ml) or spores were incubated with HEK293T cells, their internalization levels were 0.18 or 1.18 per HEK293T cells, respectively. We also examined the internalization of spore wall-derived RNA-bound latex beads (Fig. 3b) by HEK293T cells. Latex beads adsorbed spore wall-derived RNA more than tRNA for unknown reason (Fig. 3a), although no differences were found between the internalization levels of spore wall-derived RNA-bound latex beads and tRNA-bound latex beads (Fig. 3b).

Like latex beads that contain surface amine groups, $dit1\Delta$ spores have a positively charged chitosan layer at the spore wall

surface[12] (Supplementary Fig. 1a). Nevertheless, wild-type spores could hold more RNA than $dit1\Delta$ spores (Fig. 2c), suggesting the presence of machinery to accommodate RNA in the dityrosine layer. This machinery likely involves proteins, given that protease treatment resulted in a loss of spore wall-bound RNA (Supplementary Fig. 6a). The RNA-binding machinery persists on the spore wall during the high-salt wash. Indeed, high-salt-washed spores adsorbed spore wall-derived RNA (Fig. 4a). Internalization of high-salt-washed spores by HEK293T cells was improved by binding of spore wall-derived RNA (Fig. 4b). The levels of internalization of spore wall-derived RNA-bound high-salt-washed spores were higher than those of spore wall-derived RNA-bound latex beads; when the phagocytosis assay was performed with the $1.2 \times 10^7$ spore wall-derived RNA-bound spores (incubated with the RNA at $40\,\mu g/2 \times 10^8$ spores/ml) or latex beads (incubated with the RNA at $300\,\mu g/2 \times 10^8$ beads/ml), their internalization levels were 0.87 or 0.22 per HEK293T cells, respectively (Figs. 3b, 4b). Since high-salt-washed spores adsorbed more polynucleotides than latex beads (Supplementary Fig. 6b), the efficiency of internalization by HEK293T cells may be relative to the amount of polynucleotides attached to particles. As mentioned earlier, no differences were found between the internalization levels of spore wall-derived RNA- and tRNA-bound latex beads. This contradiction maybe attributable to the low levels of internalization levels of polynucleotide-bound latex beads (Fig. 3b).

Apart from spore wall-derived RNA, spores adsorbed purified tRNA and a single-stranded 22-nt DNA fragment named DNA1 (Fig. 4a). DNA1 is a mimic of a tRNA$^{Asp}$ fragment that was most abundantly found in spore wall-bound RNA. The amount of polynucleotides bound to high-salt-washed spores was diminished by protease treatment (Supplementary Fig. 6b), which is consistent with the hypothesis that proteins are involved in the polynucleotide-binding machinery. Internalization of high-salt-washed spores by HEK293T cells was improved by binding of tRNA and DNA1 (Fig. 4b). As with latex beads, spore wall-derived RNA bound to spores more efficiently than other polynucleotides (Fig. 4a). In line with the hypothesis that the levels of particle internalization by HEK293T cells were relative to the amount of polynucleotides bound to the particles, spores that had adsorbed spore wall-derived RNA were internalized more efficiently than spores that had adsorbed other polynucleotides before their bindings were saturated (Fig. 4b). Then, we analyzed the binding of another single-stranded DNA fragment which

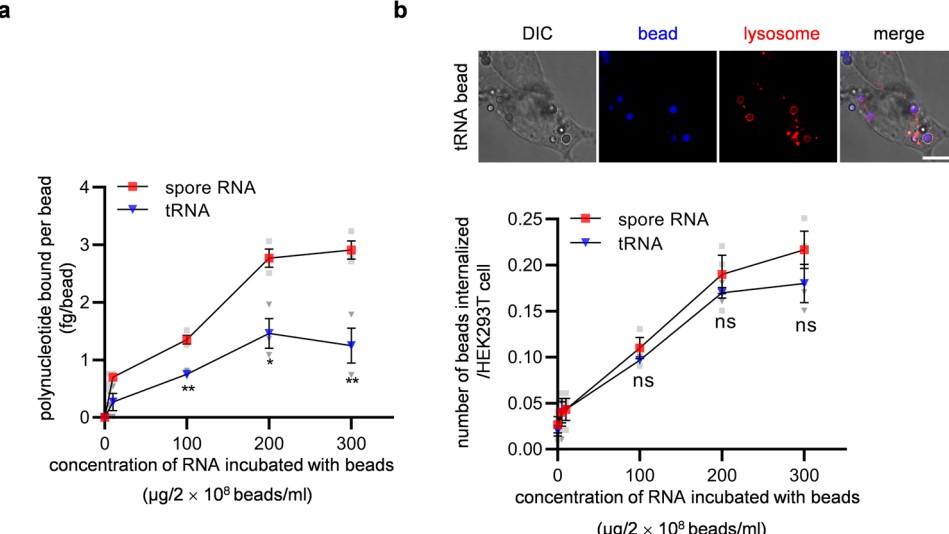

**Fig. 3 RNA induces phagocytosis by NPPs. a** Amounts of RNA bound to latex beads. Spore wall-derived RNA (spore RNA) or tRNA were incubated with latex beads at the indicated concentrations, and amounts of RNA bound to the beads were measured. **b** Internalization of RNA-bound latex beads in HEK293T cells. Upper panels: Representative images of cells with internalized tRNA-bound latex beads. Scale bar, 10 μm. Lower panel: The indicated RNA-bound latex beads were incubated with HEK293T cells at the indicated concentrations. Mean numbers of beads internalized per HEK293T cell are shown. Data were presented as the mean ± SEM. Statistical significance was determined by two-tailed unpaired Student's t-tests. $n = 3$. *$P < 0.05$; **$P < 0.01$; ns not significant ($P \geq 0.05$).

consisted of a sequence that was not related to tRNAs (its sequence is shown in Supplementary Table 1); the DNA fragment was named DNA2. DNA1 and DNA2 bound to spores at similar levels (Supplementary Fig. 6b). Notably, two DNA fragments with distinct sequences induced internalization of high-salt-washed spores at comparable levels (Fig. 4c). These results collectively suggest that polynucleotides serve as ligands to induce phagocytosis and the nucleotide sequence is not critical to the induction of internalization by cultured cells. Spores are internalized by HEK293T cells more efficiently than polynucleotide-bound latex beads maybe because spores can hold more ligand (RNA) than latex beads (Supplementary Fig. 6b). In further support of this hypothesis, a competitive inhibition assay (Fig. 4d) showed that the addition of free polynucleotides inhibited the internalization of spores by HEK293T cells.

**RAGE is a phagocytic receptor in NPPs.** Next, we sought to identify the receptor that mediates the internalization of spores in mammalian cells. Given that polynucleotides are most likely ligands, several receptors known to bind polynucleotides[18,28–32] were overexpressed as red fluorescent protein (RFP) fusions in HEK293 cells (not HEK293T cells) and assessed for their effects on spore internalization (Fig. 5a). HEK293 cells were used in this experiment because this cell line exhibited lower phagocytic efficiency than HEK293T (Fig. 1c); we speculated that phagocytic levels in HEK293 could be enhanced by overexpression of the receptor. We found that the levels of spore internalization were elevated by more than twofold by the expression of RAGE-RFP (Fig. 5a). Expression levels of *RAGE* mRNA in HEK293T cells were approximately 1.5-fold higher than those in HEK293 cells (Supplementary Fig. 7a). Spore internalization in HeLa and HEK293T cells was also enhanced by the overexpression of RAGE (Supplementary Fig. 7b). Consistent with a role for RAGE in spore internalization, this process was decreased 10-fold in *RAGE* knockout cells (Fig. 5b). The loss of RAGE expression in HEK293T *RAGE* KO cells was verified by western blotting analysis (Supplementary Fig. 7c). The result of FACS analysis also showed that phagocytosis of spores by HEK293T was decreased

by *RAGE* KO (Supplementary Fig. 7d). Furthermore, the defect in spore internalization in HEK293T *RAGE* KO cells was rescued by the expression of RAGE-RFP (Fig. 5b). Internalization of polynucleotide-bound beads was also decreased in HEK293T *RAGE* KO cells (Fig. 5c). In addition to RAGE, we found that internalization of spores in HEK293 cells was also increased by overexpression of Toll-like receptor 3 (TLR3) (Fig. 5a). However, the levels of spore internalization in HEK293T cells were not altered by *TLR3* knockout (Supplementary Fig. 7e), showing that this receptor is either not required for spore internalization or is redundant with another receptor providing the same function. Therefore, TLR3 is not considered further here.

RAGE has two polynucleotide-binding patches, which are termed Site 1 and Site 2[18]. Spore internalization in HEK293T *RAGE* KO cells expressing RAGE-RFPs harboring mutations in either of the nucleotide-binding sites (termed RAGE^mut1-RFP and RAGE^mut2-RFP, respectively) was lower than that seen in cells expressing RAGE-RFP (Fig. 5b). To further analyze the binding between RAGE and RNA, we performed in vitro binding assays. The extracellular region of RAGE (amino acids 1–341) fused to FLAG and His tags was expressed in HEK293 cells; the truncated RAGE was named RAGE^1–341-His-FLAG. Purified RAGE^1–341-His-FLAG was attached to agarose beads conjugated with an anti-FLAG tag antibody and incubated with an RNA fragment (a mimic of DNA1) fused to cyanine 3 (Cy3). As shown in Fig. 5d, the RNA fragment is bound to the extracellular region of RAGE. Furthermore, we found that RAGE^1–341-His-FLAG was precipitated with tRNA-bound latex beads more than bare latex beads (Fig. 5e). These results demonstrate that RNA directly binds to RAGE. Predicted molecular weights of RAGE^1–341-His-FLAG is 40.7 kDa. Purified RAGE^1–341-His-FLAG (Fig. 5d) detected by SDS-PAGE was larger than the predicted molecular weight (Fig. 5d), presumably because RAGE has two N-linked glycosylation sites.

To assess the localization of RAGE during the internalization of spores, we performed time-lapse microscopy. This analysis showed that the spore was engulfed by green fluorescent protein (GFP) fused to RAGE before its internalization (Fig. 5f and Supplementary Video 1). RAGE-GFP was internalized together

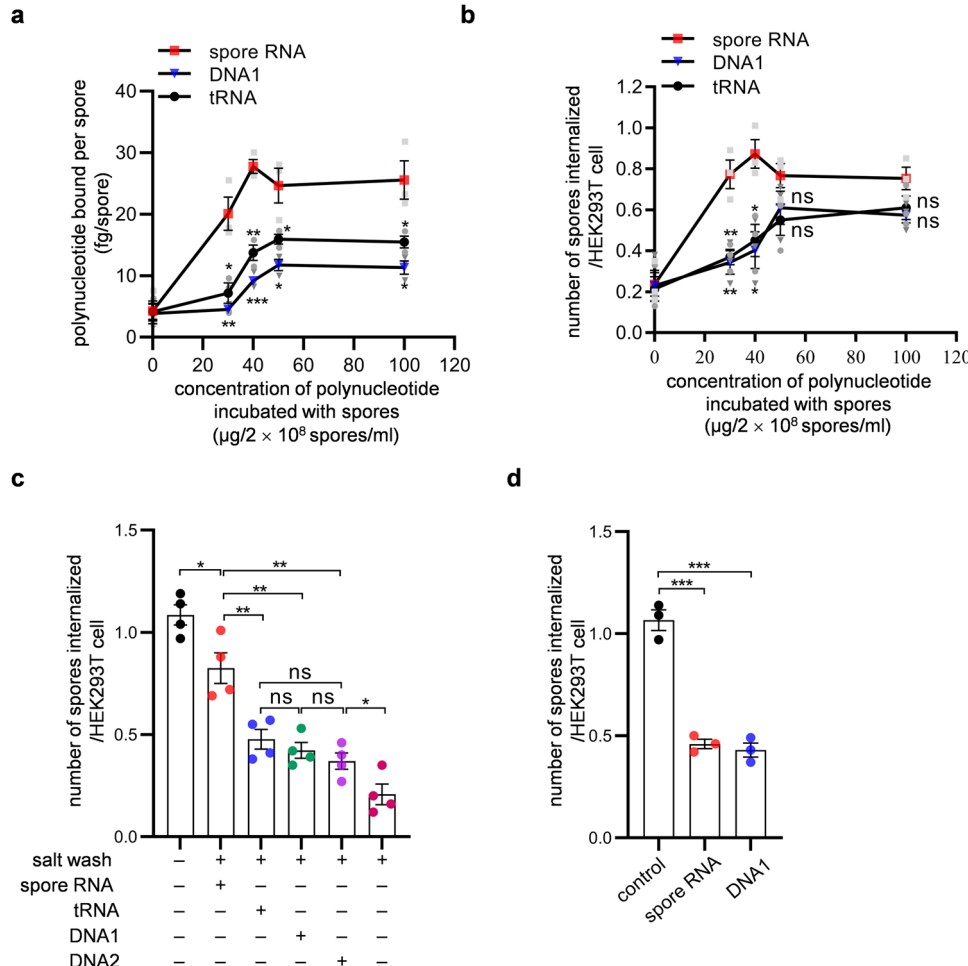

**Fig. 4 Internalization of spores in HEK293T cells is induced by polynucleotides. a** Amounts of polynucleotides bound to high-salt-washed spores. High-salt-washed spores were incubated with spore wall-derived RNA (spore RNA), tRNA, or DNA1 at the indicated concentrations. Polynucleotides bound to the spores were eluted in a high-salt solution and their amounts were measured. **b** Internalization of polynucleotide-bound high-salt-washed spores in HEK293T cells. High-salt-washed spores incubated with spore wall-derived RNA (spore RNA), tRNA, or DNA1 at the indicated concentrations were incubated with HEK293T cells. Mean numbers of spores internalized per HEK293T cell are shown. **c** Internalization of high-salt-washed spores ($2 \times 10^8$) incubated with (+) or without (−) 40 μg of indicated polynucleotides in HEK293T cell. As a control, untreated spores were incubated with HEK293T cells (salt wash −). **d** Spore internalization in HEK293T cells in the presence or absence (control) of the indicated polynucleotides. Data were presented as the mean ± SEM. Statistical significance was determined by two-tailed unpaired Student's $t$-tests. For **a**, **b**, Student's $t$-tests was performed between spore wall-derived RNA and tRNA or DNA1 data. $n = 3$ (**a**, **b**, **d**), $n = 4$ (**c**). *$P < 0.05$; **$P < 0.01$; ***$P < 0.001$; ns not significant ($P \geq 0.05$).

with the spores, and later, the GFP fusion- and spore-containing compartments were stained with LysoTracker (Fig. 5f and Supplementary Video 1). Levels of fluorescent intensities of RFP in certain areas in phagocytic cups (plasma membrane engulfing spores) were higher than those of areas adjacent to the phagocytic cup in the plasma membrane in fixed cells (Supplementary Fig. 7f, g). This result suggests that RAGE-RFP is recruited to phagocytic cups. Taken together, our findings indicated that spores are internalized via RAGE-mediated phagocytosis.

RAGE is primarily expressed in the lung compared to other organs[33,34]. To assess whether RAGE-mediated phagocytosis occurs in vivo, a spore internalization assay was performed with primary mouse alveolar type II (ATII) epithelial cells. Spores were internalized by primary ATII cells (Fig. 6a, b). Internalization of spores in ATII cells was inhibited by reagents or mutations that similarly compromised spore internalization in HEK293T cells (Fig. 6b–d). To determine whether Rage (the mouse homolog) is required for spore internalization in ATII cells, RNA interference (RNAi)-mediated Rage knockdown was performed. Quantitative

real-time PCR (qRT-PCR) analysis showed that the levels of the mouse Rage transcript were decreased by transfection of ATII cells with small interfering RNAs (siRNAs) targeting Rage (Fig. 6e, left panel). Accordingly, the levels of spore internalization by the mouse cells were decreased in Rage knockdown cells relative to control ATII cells (Fig. 6e, right panel). This result suggests that RAGE-mediated phagocytosis occurs in various NPPs.

**Various molecules are internalized via RAGE-mediated phagocytosis in NPPs.** Since RAGE is a multiligand receptor, we examined whether RAGE-mediated phagocytosis is induced by another ligand, HMGB1. HMGB1-bound latex beads (HMGB1 beads) were prepared by crosslinking recombinant HMGB1 to latex beads. Internalization of latex beads was increased threefold by the binding of HMGB1 to the beads (Fig. 7a, b and Supplementary Fig. 8a, b). Compared to wild-type HEK293T cells, the internalization levels of HMGB1 beads were decreased in RAGE KO cells (Fig. 7b). These results suggest that HMGB1 induces phagocytosis by NPPs.

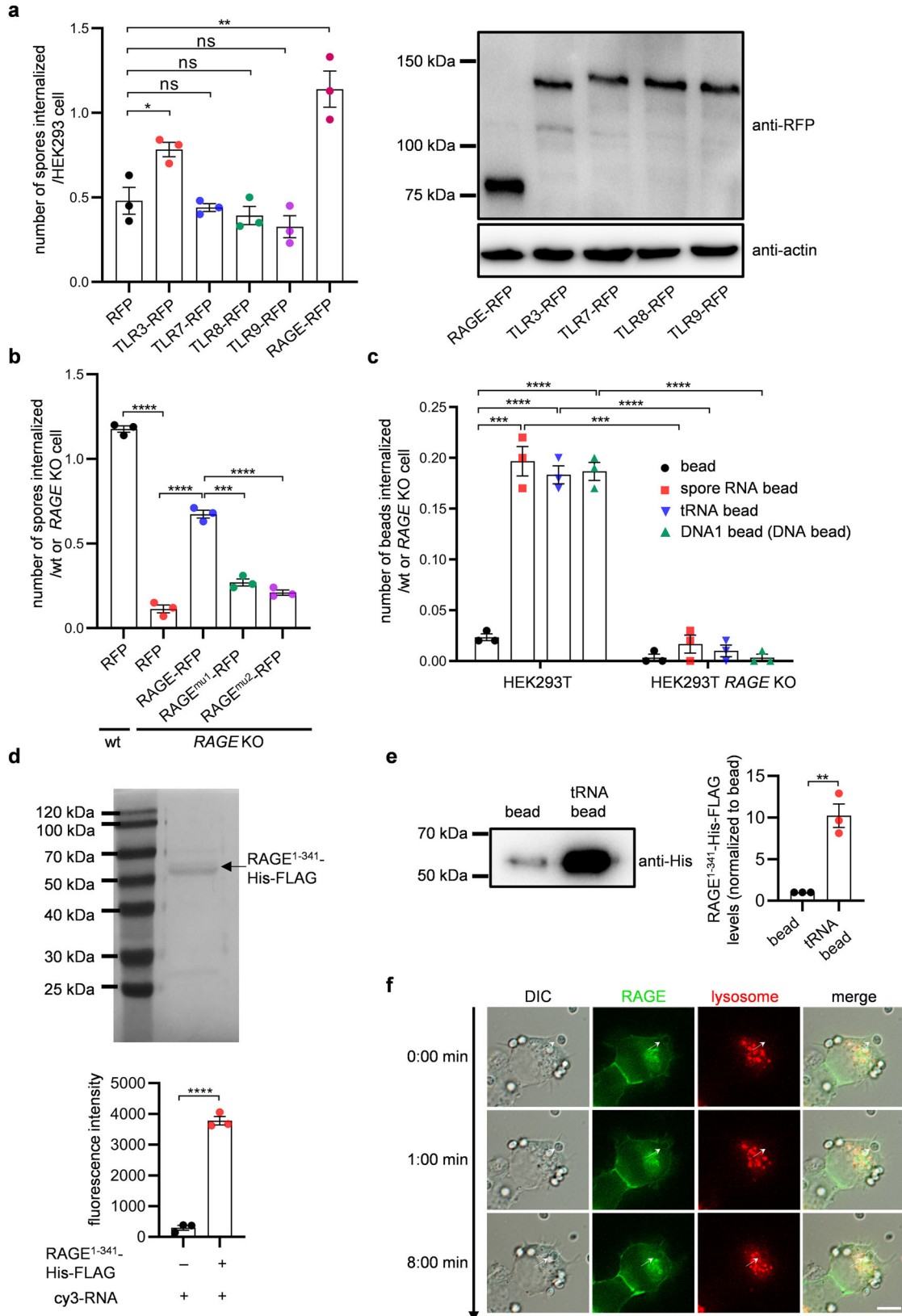

Like polynucleotide-bound latex beads, the levels of internalization of HMGB1-bound beads were lower than those of spores (compare Fig. 2a (control) and Fig. 7b (DNA and HMGB1)). As described earlier, this difference may indicate that spores display greater densities of ligand (RNA) than latex beads (Supplementary Fig. 6b). Additionally, or alternatively, the spore wall may contain other substance(s) that can augment or actively induce internalization. Regarding the latter possibility, a previous study showed that the RAGE-mediated inflammatory response is activated synergistically by DNA and HMGB1[35]. By analogy, we found that internalization of DNA- and HMGB1-bound (double-labeled) beads (DNA/HMGB1 beads) by HEK293T cells was

**Fig. 5 RAGE is the receptor for internalization of spores by NPPs. a** Left panel: Internalization of spores in HEK293 cells expressing the indicated RFP fusions or RFP alone. Right panel: Western blotting analysis using an anti-RFP antibody to verify the expression of the RFP fusions. Actin is used as a loading control. **b** Internalization of spores in HEK293T (wt) or HEK293T *RAGE* KO cells expressing the indicated RFP fusions. **c** Internalization of latex beads in wild-type HEK293T or HEK293T *RAGE* KO cells. The indicated polynucleotide-bound latex beads were incubated with wild-type HEK293T or HEK293T *RAGE* KO cells. **d** Top panel: Purified RAGE$^{1-341}$-His-FLAG was subjected to SDS-PAGE analysis and stained with Coomassie brilliant blue. Bottom panel: Quantification of fluorescence intensities of Cy3-RNA precipitated with RAGE$^{1-341}$-His-FLAG attached to agarose beads or bare agarose beads (control). **e** Left panel: Binding of RAGE$^{1-341}$-His-FLAG to tRNA-bound latex beads. RAGE$^{1-341}$-His-FLAG was incubated with tRNA-bound beads or bare beads (bead), and the protein precipitated with the beads were detected with western blotting analysis using an anti-His tag antibody. Right panel: Quantification of RAGE$^{1-341}$-His-FLAG precipitated with tRNA-bound beads or bare beads (bead). Relative intensities of RAGE$^{1-341}$-His-FLAG precipitated with the beads were shown. The intensity of RAGE$^{1-341}$-His-FLAG precipitated with bare beads was defined as 1. **f** The internalization process of spores in HEK293T cells followed by time-lapse microscopy. HEK293T cells transiently expressing RAGE-GFP were preincubated with LysoTracker Red (lysosome) and were then incubated with spores. Representative images of a spore being internalized (arrow) are shown. Scale bar, 10 µm. Data were presented as the mean ± SEM (**a–e**). Statistical significance was determined by two-tailed unpaired Student's *t*-tests. $n = 3$. *$P < 0.05$; **$P < 0.01$; ***$P < 0.001$; ****$P < 0.0001$; ns not significant ($P \geq 0.05$).

higher than that observed with DNA or HMGB1 beads (i.e., single-labeled particles) (Fig. 7a, b). For this assay, DNA/HMGB1 beads were prepared by adsorbing HMGB1 to DNA beads. In DNA/HMGB1 beads, HMGB1 appeared to bind to DNA on latex beads, given that HMGB1 was not adsorbed by bare beads (Supplementary Fig. 8c). The levels of DNA bead internalization by HEK293T cells were increased by the binding of HMGB1 (Supplementary Fig. 8c, d). The levels of DNA/HMGB1 bead internalization by HEK293T *RAGE* KO cells were lower (Fig. 7b). Thus, DNA and HMGB1 synergistically augment RAGE-mediated phagocytosis.

Then, we performed a similar experiment using another DNA-binding protein, histone. Extracellular histones are known to act as DAMPs[36], but they previously have been reported not to serve as ligands for RAGE. Thus, first, we utilized histones as a negative control. However, histone-labeled beads were internalized in HEK293T cells to levels approximately twofold higher than DNA/HMGB1 beads (Fig. 7a, b and Supplementary Fig. 9a, b). In contrast, bovine serum albumin (BSA)-labeled beads were not internalized in HEK293T cells (Fig. 7b), showing that RAGE-mediated phagocytosis is not induced by proteins in general. Notably, the internalization of histone beads was decreased by *RAGE* knockout (Fig. 7b). These results suggest that histones are another ligand for RAGE. In support of this hypothesis, RAGE-GFP was precipitated with histone beads, but not with BSA beads, from a cell lysate harboring the RAGE-GFP fusion protein (Supplementary Fig. 9c). As seen with HMGB1, internalization of DNA beads was improved by the additional binding of histones (Fig. 7a, b and Supplementary Fig. 9d, e). In HEK293T *RAGE* KO cells, internalization of the DNA/histone beads was decreased (Fig. 7b). Histone beads and DNA/histone beads were also internalized by mouse ATII cells (Fig. 7c). We note, however, that internalization of beads by ATII cells was not induced by HMGB1 (Fig. 7c). Furthermore, unlike HEK293T cells, primary mouse ATII cells were more efficient at internalizing spores than any of the modified artificial beads (compare Fig. 6c (control) and Fig. 7c). Nevertheless, internalization of DNA/histone beads or histone beads by ATII cells was diminished by knockdown of *Rage* (Fig. 7d), showing that internalization of the beads by these primary mouse cells is mediated by Rage-dependent phagocytosis.

Finally, we examined whether RAGE is required for HEK293T cells to internalize apoptotic cells, which are well-known targets for phagocytosis by NPPs in vivo[2]. Jurkat cells were internalized in HEK293T cells when they were treated with the apoptosis inducer staurosporine (Fig. 8a, b). In HEK293T *RAGE* KO cells, however, the levels of internalization of staurosporine-treated Jurkat cells were approximately fourfold lower than those in wild-type cells (Fig. 8b). Furthermore, we found that the sizes of Jurkat cells (fragments) internalized in HEK293T *RAGE* KO cells

were smaller than those internalized in wild-type cells (Fig. 8c). These results demonstrate that efferocytosis by NPPs is also mediated by a RAGE-dependent pathway.

## Discussion

Phagocytosis occurs in NPPs[3]; however, the mechanism is not clear. Here, we demonstrate that phagocytosis by NPPs is mediated by RAGE. Given that *RAGE* knockout cells exhibit residual activity to internalize microsized particles, NPPs may be equipped with multiple phagocytic pathways. Nevertheless, NPPs can phagocytose various molecules via the RAGE-mediated pathway due to the multiligand-recognition property of the receptor.

In vivo, NPPs are known to perform efferocytosis[2]. We show that RAGE can mediate the clearance of apoptotic corpses; presumably, PS on apoptotic cells serves as a ligand for this process. Apart from cell corpses, targets for phagocytosis by NPPs in vivo were not clear. Our results suggest that various macromolecules containing RAGE ligands are phagocytosed by NPPs. Macromolecules containing RAGE ligands are known to be present in the extracellular space[37,38]. For example, DNA fragments bound to histones or HMGB1 can be released from damaged cells. These molecules are potential targets for RAGE-mediated phagocytosis. Intriguingly, most, if not all, RAGE ligands are harmful molecules[39,40]. However, the underlying mechanism for how RAGE can recognize multiple harmful ligands awaits further structural analysis.

Direct binding between RAGE and its ligands has been demonstrated in previous reports[28,35]; in RAGE-mediated phagocytosis, their binding most likely induces the signaling pathway. We found that the internalization of polynucleotide-bound latex beads was augmented by the binding of other DNA-binding RAGE ligands. The synergistic effect may reflect the fact that the DNA/HMGB1 complex binds to RAGE with higher affinity than DNA or HMGB1 alone, as has been reported previously[35]. Our pharmacological analysis suggests that SYK and PI3K are involved in the signaling pathway. These kinases are known as downstream signaling molecules in Fcγ receptor-mediated phagocytosis[1], indicating similarity between professional and nonprofessional phagocytic pathways. The Fcγ receptor has an immunoreceptor tyrosine-based activation (ITAM) motif in the cytosolic domain[1]. SYK directly binds to the activated Fcγ receptor via the phosphorylated ITAM motif. However, unlike the Fcγ receptor, the ITAM motif is absent in the cytosolic domain of RAGE. Thus, activation of RAGE-mediated phagocytic signaling may involve an adapter protein or coreceptor. Notably, HEK293T and mouse primary ATII cells internalize particles with distinctive preferences; HEK293T cells phagocytose DNA/histone beads most efficiently, whereas ATII cells prefer spores. This result suggests that internalization efficiency is not determined

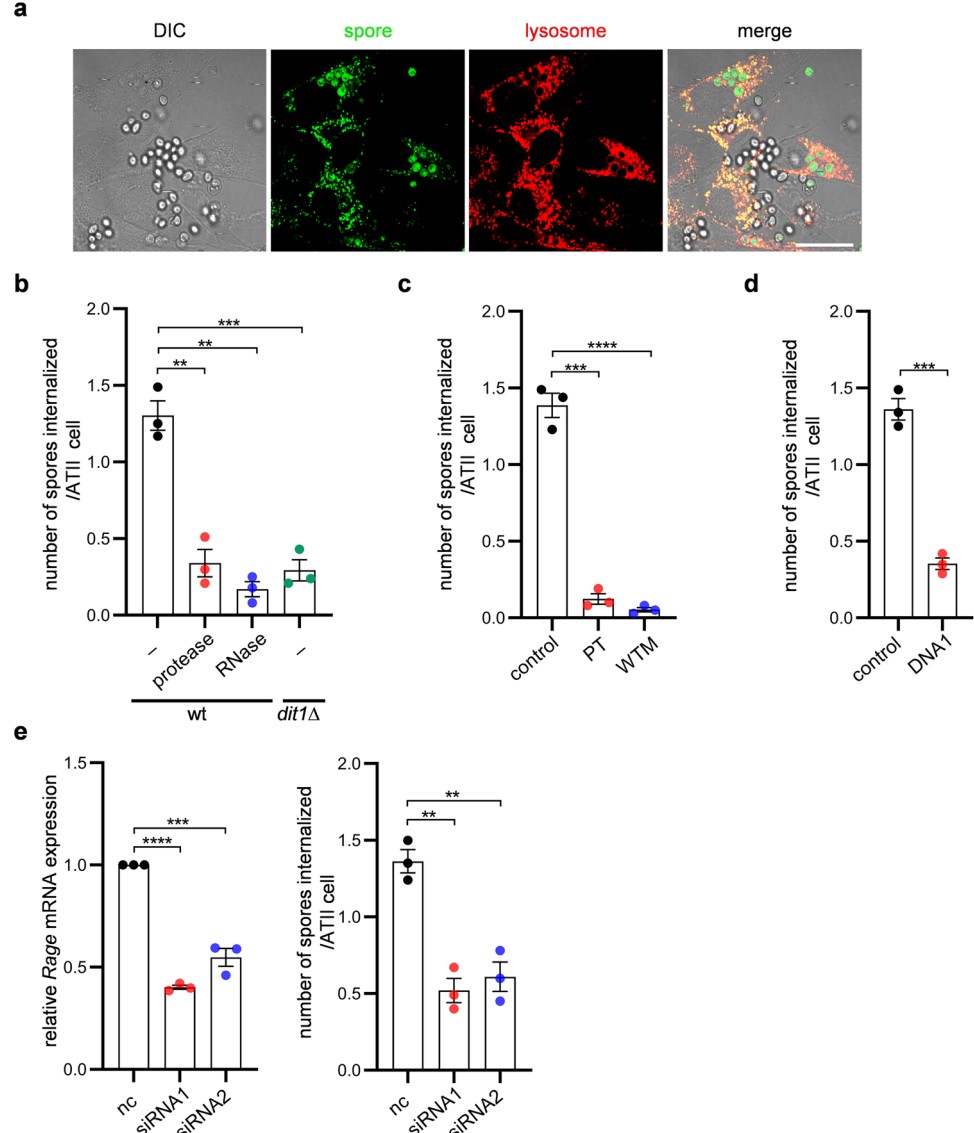

**Fig. 6 Spores are internalized in primary mouse ATII cells. a** Representative images of ATII cells with internalized spores. Images were obtained with differential interference contrast (DIC) or fluorescence microscopy (spore and lysosome). Lysosomes were stained with LysoTracker Red. Scale bar, 20 μm. **b** Internalization of wild-type (wt) or *dit1Δ* spores in ATII cells. Wild-type (wt) or *dit1Δ* spores without any treatment (−), wild-type spores treated with protease or RNase A (RNase) were incubated with ATII cells. **c** Internalization of spores in ATII cells treated with or without (control) piceatannol (PT) or wortmannin (WTM). **d** Spore internalization in ATII cells in the presence or absence (control) of DNA1. **e** Left panel: Levels of the *Rage* mRNA in primary mouse ATII cells transfected with two *Rage*-targeting siRNAs (siRNA1 and siRNA2) or a nontargeting siRNA control (nc). The ratios of *Rage* mRNA levels in cells transfected with *Rage*-targeting siRNAs to those in cells transfected with nc are shown. Right panel: Internalization of spores in *Rage* knockdown- and nc-transfected ATII cells. Data were presented as the mean ± SEM (**b**–**e**). Statistical significance was determined by two-tailed unpaired Student's t-tests. $n = 3$. **$P < 0.01$; ***$P < 0.001$; ****$P < 0.0001$.

solely by RAGE's affinity for particles. Thus, internalization efficiency may be mediated by one or more additional coreceptors.

While we focus on the mechanism of phagocytosis by NPPs in the present report, our study also provides several intriguing findings from the perspective of microbiology. In particular, we found that RNA fragments bind to the spore wall. The spore wall is equipped with machinery to hold RNA, suggesting that RNA fragments may be beneficial for spores. Since *S. cerevisiae* is a nonpathogenic yeast, it is unlikely that spore wall-bound RNA is invasive machinery. Apart from the spore wall, the presence of extracellular RNA or DNA is reported in several biological structures, including neutrophil extracellular traps and biofilms[41,42]. In these structures, extracellular polynucleotides are

used as materials to protect cells or organisms. By analogy, we propose that spore wall-bound RNA has a protective function. Given that spores are formed in the cytosol, it would be reasonable to use otherwise useless RNA fragments as protective materials. Thus, it would be intriguing to assess whether RNA is included in the spore wall of other organisms. In the yeast spore wall, tRNA, especially tRNA$^{Asp}$, fragments are concentrated in spore wall-bound RNA for unknown reasons. In mammalian cell cultures and biofluids, fragments of tRNA$^{Gly}$ and tRNA$^{Glu}$ are major nonvesicular extracellular RNAs because they adopt RNase-resistant forms[43]. In yeast, therefore, tRNA$^{Asp}$ may be resistant to RNase digestion in the ascal cytosol. Alternatively, the bias may be attributable to the binding specificity of the RNA-binding machinery present in the spore wall.

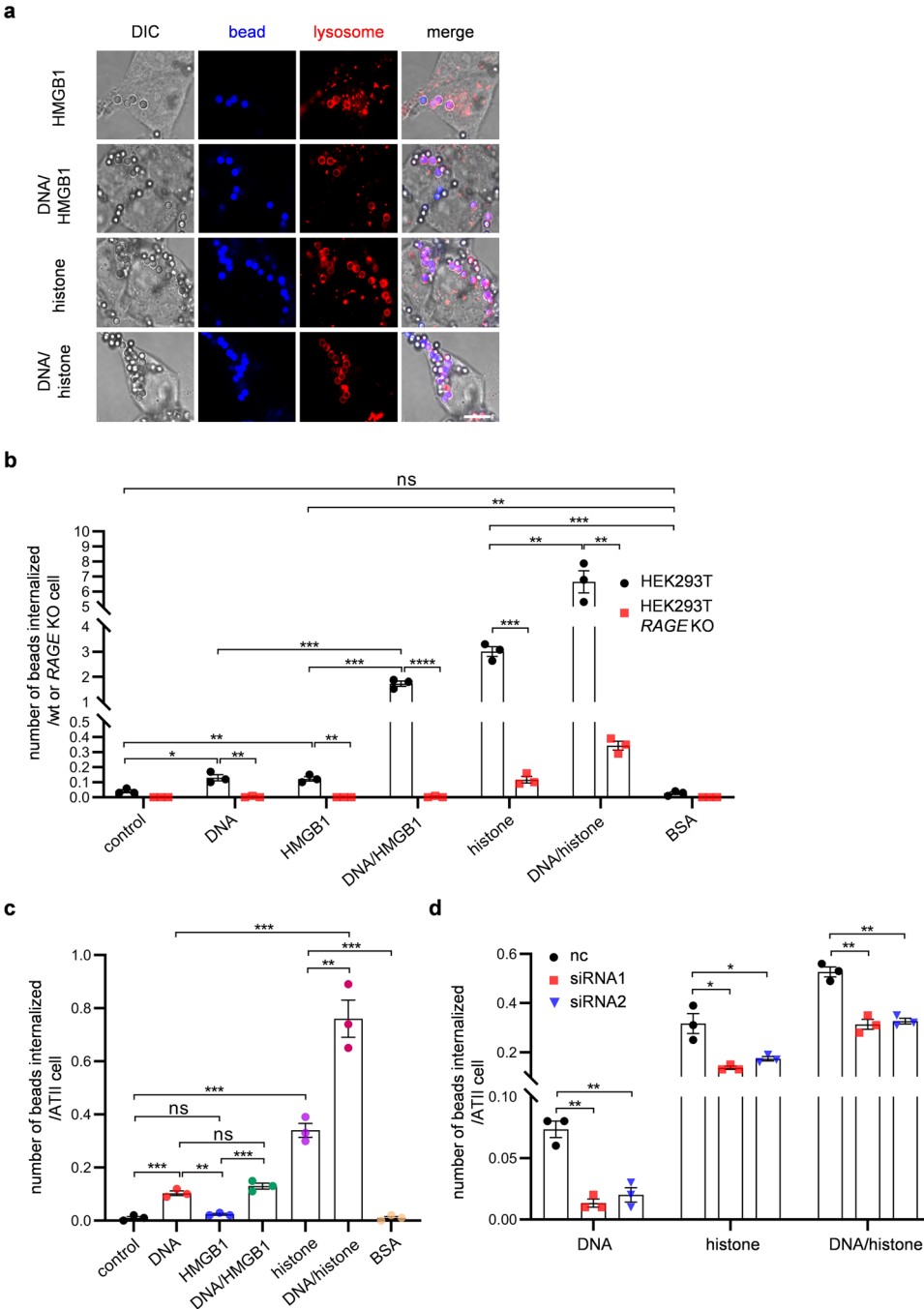

**Fig. 7 RAGE ligands induce phagocytosis in NPPs. a** Representative images of cells with internalized latex beads. For HMGB1 or histone beads, 49.57 or 44.40 fg/bead (respectively) of the proteins were bound to latex beads. For DNA/HMGB1 or DNA/histone beads, 65.94 or 61.39 fg/bead (respectively) of the proteins were bound to DNA beads (amount of DNA1 bound was 1.27 fg/bead). Images were obtained with differential interference contrast (DIC) or fluorescence microscopy (spore and lysosome). Lysosomes were stained with LysoTracker Red. Scale bar, 10 μm. **b** Internalization of latex beads in HEK293T or HEK293T *RAGE* KO cells. Bare latex beads (control) or the indicated beads were incubated with HEK293T or HEK293T *RAGE* KO cells. For BSA beads, 48.02 fg/bead of the protein-bound latex beads. **c** Internalization of latex beads in ATII cells. The indicated types of latex beads were incubated with primary mouse ATII cells. **d** Internalization of latex beads in ATII cells transfected with *Rage*-targeting siRNAs. ATII cells were transfected with two *Rage*-targeting siRNAs (siRNA1 and siRNA2) or a nontargeting siRNA control (nc). These cells were incubated with the indicated beads as described in **a**. Data were presented as the mean ± SEM (**b–d**). Statistical significance was determined by two-tailed unpaired Student's *t*-tests. n = 3 (**b–d**). *P < 0.05; **P < 0.01; ***P < 0.001; ****P < 0.0001; ns not significant (P ≥ 0.05).

Yeast spores exhibit unique properties in that they are internalized by NPPs far more efficiently than polynucleotide-bound latex beads. There are two possibilities to explain the efficient phagocytosis of spores. One is that spores could accommodate more polynucleotides than latex beads with surface amine groups. The other possibility is that additional molecules bound to the RNA on the spore wall could improve internalization efficiency, although currently, the identity of such a molecule remains elusive. The RNA-binding machinery in the spore wall would be intriguing not only from the point of view of microbiology but also for the development of delivery systems for mammalian cells.

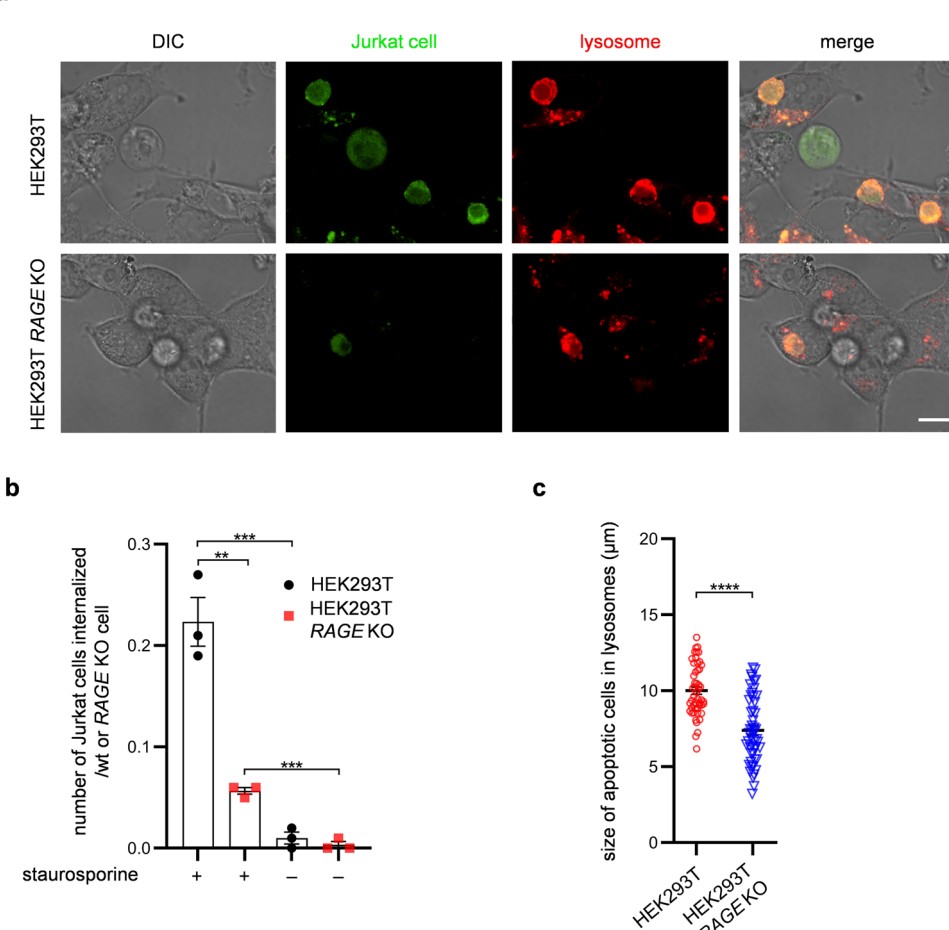

**Fig. 8 RAGE is required for the internalization of apoptotic Jurkat cells in HEK293T cells. a** Representative images of HEK293T or HEK293T *RAGE* KO cells with internalized apoptotic Jurkat cells expressing GFP obtained with differential interference contrast (DIC) or fluorescence microscopy (apoptotic cell and lysosome). Lysosomes were stained with LysoTracker Red. Scale bar, 10 μm. **b** Internalization of apoptotic Jurkat cells in HEK293T or HEK293T *RAGE* KO cells. Apoptotic cells were incubated with HEK293T or HEK293T *RAGE* KO cells at $10^6$ Jurkat cells/4–6 × $10^5$ wild-type or *RAGE* KO cells/ml. $n = 3$. **c** Distribution of size of Jurkat cell (fragment) internalized in HEK293T ($n = 50$) or HEK293T *RAGE* KO ($n = 50$) cells. The size was determined by measuring the longest dimension of internalized cells. Data were presented as the mean ± SEM (**b**). Statistical significance was determined by two-tailed unpaired Student's $t$-tests (**b**, **c**). **$P < 0.01$; ***$P < 0.001$; ****$P < 0.0001$.

RAGE is known to be involved in a number of disorders[44–46]. Activation of RAGE can induce inflammatory responses, and many disorders are reported to be linked to RAGE-mediated chronic inflammation. Nevertheless, some disorders, such as infectious diseases, may be related to RAGE's function as a phagocytic receptor. RAGE is highly expressed in the lung, where epithelial cells encounter a variety of microorganisms[47]. Given that yeast spores are phagocytosed in NPPs, other microorganisms may be internalized in epithelial tissues via RAGE-mediated phagocytosis. In this context, it is notable that various bacteria form biofilms containing polynucleotides. Implications for RAGE in infectious diseases have been reported in mouse models. Intriguingly, RAGE may cause either beneficial or detrimental effects depending on the pathological organisms and conditions[44]. Variable results may be attributable to the pathogen's tolerance of endocytic compartments in NPPs; for some pathogens, the phagocytic process would be helpful for host invasion. In support of this hypothesis, spores internalized in HEK293T cells are alive and commence germination, which leads to the death of the cultured cells.

Since NPPs are not equipped with microbicidal mechanisms, they are vulnerable to pathogens compared to professional phagocytes. Thus, professional phagocytes perform better for the clearance of pathogens. Notably, a previous study showed that phagocytic activity in NPPs is suppressed by IGF-1 released from professional phagocytes[48]. This cross-regulation may be beneficial to prevent pathogen propagation. NPPs and professional phagocytes may interact in various ways and contribute to the development and maintenance of tissues and to the disease response. We hope that our findings will pave the way for further analyses and insights into the physiological roles of phagocytosis by NPPs.

## Materials and methods

**Mammalian cells**. HEK293T, HEK293, HeLa, and HUC cells were obtained from American Type Culture Collection (ATCC). These cells were cultured in Dulbecco's modified Eagle's medium (Biological Industries) containing 10% (v/v) fetal calf serum (FCS) (Biological Industries). The cells were maintained at 37 °C in a humidified atmosphere with 5% $CO_2$. Mouse primary ATII cells were purchased from Procell and cultured in ATII cell special medium (Procell). Jurkat cells were obtained from ATCC and cultured in Roswell Park Memorial Institute (RPMI) 1640 (Biological Industries) containing 10% (v/v) FCS. FCS was inactivated by incubation at 56 °C for 45 min before addition to media. Jurkat cells stably expressing GFP were cultured in RPMI 1640 containing 10% (v/v) FCS and 5 μg/ml blasticidin (InvivoGen). The cells were maintained at 37 °C in a humidified atmosphere with 5% $CO_2$.

**Yeast strains, culture, and spore preparation**. The yeast strains used in this study are AN120 (wild-type) (*MATα/MATa ARG4/arg4-NspI his3ΔSK/his3ΔSK ho::LYS2/ho::LYS2 leu2/leu2 lys2/lys2 RME1/rme1::LEU2 trp1::hisG/trp1::hisG ura3/ura3*)[49] and HW3 (*dit1Δ*) (*MATα/MATa ARG4/arg4-NspI his3ΔSK/his3ΔSK ho::LYS2/ho::LYS2 leu2/leu2 lys2/lys2 RME1/rme1::LEU2 trp1::hisG/trp1::hisG ura3/ura3 dit1Δ::his5⁺/dit1Δ::his5⁺*)[50]. These strains are SK-1 strain background that sporulates with high efficiency[51]. To express GFP in AN120, the haploid strains[49] were transformed with pRS306TEF-GFP digested with *Eco*RV and the transformants were mated.

Yeast cells were cultured in YPAD (10 g/l yeast extract, 20 g/l peptone, 30 mg/l adenine, and 20 g/l glucose). Agar (20 g/l) was added to prepare plates. Yeast cells were sporulated as previously described[50]. Briefly, yeast cells derived from a single colony were cultured overnight in 5 ml of YPAD liquid medium. A 0.1 ml aliquot of the culture was transferred into 5 ml of YPAcatate (10 g/l yeast extract, 20 g/l peptone, and 20 g/l potassium acetate) and grown overnight. The cells were harvested by centrifugation, resuspended in 2% potassium acetate medium at a concentration of $3 \times 10^7$ cells/ml, and cultured for 24 h.

To release spores from asci, the asci were suspended in 5 ml of lyticase buffer and 20 μl of lyticase (Sigma-Aldrich) stock solution (10,000 U/ml was dissolved in 500 μl of 50% glycerol) was added. After a 3-h incubation at 37 °C, the spores were washed twice with lyticase buffer. Then, the spores were resuspended in water and sonicated with an ultrasonic disruptor (Xinchen Biological Technology) to disrupt the asci membrane. The sonication conditions were as follows: power, 45%; duration, 20 min with cycles of 5 s on and 2 s off. The spores were then washed three times with 0.5% Tween-20 and two times with water. The number of spores were counted with a platelet counter.

Spores washed with high salt were prepared by incubation of the spores in 0.6 M NaCl solution for 30 s and then washed twice with water. To treat the spores with RNase A or DNase I, $2 \times 10^7$ spores were incubated with 3 U of RNase A (TaKaRa) or 3 U DNase I (Beyotime) in water at 37 °C for 30 min. The spores were washed twice with water. To prepare proteinase-treated spores, $2 \times 10^7$ spores were incubated with 2 U of proteinase (Sigma-Aldrich) in water at 37 °C for 30 min and washed twice with water.

**Plasmids**. All the oligonucleotides and plasmids used in this study are listed in Supplementary Tables 1 and 2, respectively. For the CRISPR-Cas9 system used to knockout target genes, guide RNA sequences were designed using the E-CRISP website[52] (RRID:SCR_019088), and the corresponding DNA fragments were ligated into the *Bpi*I-digested vector pX330EGFP-hU6-gRNA-hSpCas9[53]. The *RAGE* and *SYK* cDNA fragments were amplified from cDNA derived from HEK293T cells and cloned into the pME-tagRFP or pME-mEGFP vectors[54] to generate pME-RAGE-tagRFP, pME-RAGE-mEGFP, and pME-SYK-mEGFP. DNA-binding site RAGE mutants were constructed with PCR-based site-directed mutagenesis. pME-Hyg-3FLAG[54] was used to clone RAGE$^{1-341}$-His-FLAG. The sequences of the primers used to introduce the mutations are listed in Supplementary Table 1. The *TLR3* cDNA fragment was amplified from human cDNA derived from HEK293T cells. *TLR7* and *TLR8* cDNA fragments were amplified from human cDNA derived from THP1 cells. *TLR9* was amplified from an Ultimate™ human ORF clone (clone #, IOH21944). To construct the yeast GFP expression plasmid pRS306TEF-GFP, GFP was amplified by PCR using pFA6a-GFP (S65T)-HIS3MX6[55] as a template. The *TEF2* promoter and *CYC1* terminator were obtained from pRS316TEF[56]. pLIB2-mEGFP-BSD was used to make Jurkat cells stably expressing GFP. The GFP (mEGFP) was digested out of pME-mEGFP.

**Transfection of mammalian cells**. For plasmid DNA transfection, cells were grown to 60% confluence in 24-well slides and transfected with plasmid DNA (0.8 μg/well) using Lipofectamine™ 2000 (Thermo Fisher Scientific) according to the manufacturer's instructions.

Retrovirus-based transfection[57] was used to construct Jurkat cells stably expressing GFP. HEK293T cells ($10^6$) were transfected with 1 μg pGP, 1 μg pLC-VSVG, and 2 μg pLIB2-mEGFP-BSD using Lipofectamine™ 2000 according to the manufacturer's instructions. After 36 h of incubation, the medium was filtered with a 0.22 μm filter and mixed with the same amount of DMEM supplemented with 16 mg/ml hexadimethrine bromide (Sigma). The medium containing retrovirus was incubated with Jurkat cells overnight. After 5 days of culture, the cells expressing GFP were sorted using a cell sorter S3e (Bio-Rad).

**Viability analysis of cultured cells**. A round glass coverslip (14 mm in diameter) was placed on the bottom of 24-well plates. Cells ($1.5 \times 10^5$/well) were seeded in the plate and allowed to grow to a density of $2-3 \times 10^5$/well. The cells were incubated with $0.6 \times 10^7$ spores for 12 h at 37 °C in a humidified atmosphere with 5% $CO_2$. After being washed twice with PBS (Sangon Biotech), the cells were stained with 10 μM propidium iodide for 15 min at room temperature. After being washed with PBS three times, the dead cells were counted under a fluorescence microscope.

**Establishment of knockout cell lines**. To generate the *RAGE* knockout cell line, HEK293T cells were transiently transfected with two plasmids, pX330-RAGE gRNA1 and pX330-RAGE gRNA2 (Supplementary Table 2), which carried the following gRNAs targeting the exon regions of the *RAGE* gene: pX330-RAGE

gRNA1, CAAGAAACCACCCCAGCGGC; pX330-RAGE gRNA2, TCTTACGG TAGACACGGACT. After 3 days of incubation, the cells expressing EGFP were sorted using a cell sorter S3e (Bio-Rad). Then, the collected cells were cultured for 8 days and subjected to limiting dilution to obtain *RAGE*-knockout clone candidates. Clones lacking the wild-type alleles of the target region were selected by PCR using the RAGE check F1 and RAGE check R1 primers (Supplementary Table 1), and positive clones were further verified with DNA sequences using the Sanger method. The *TLR3*-knockout cell line was similarly generated using pX330-TLR3 gRNA1 and pX330-TLR3 gRNA2. The sequences of the gRNAs ligated into pX330-TLR3 gRNA1 and pX330-TLR3 gRNA2 are GTACCTGAGTCAACTTCAGG and GCCTTGTATCTACTTTTGGG, respectively. PCR primers used to select clones lacking the wild-type alleles of the target region were TLR3 check F1 and TLR3 check R1 (Supplementary Table 1). The *SYK*-knockout cell line was similarly generated using pX330-SYK gRNA1 and pX330-SYK gRNA2. The sequences of the gRNAs ligated into pX330-SYK gRNA1 and pX330-SYK gRNA2 are GGCA-GAAGATTACCTGGTCC and GCTTCTTGAGGAGGCAGACC, respectively. PCR primers used to select clones lacking the wild-type alleles of the target region were SYK check F1 and SYK check R1 (Supplementary Table 1).

**Gene knockdown experiment**. Knockdown of *Rage* in mouse ATII cells was performed with RNA interference (RNAi) with small interfering RNA (siRNA). One siRNA (siRNA1) was purchased from Santa Cruz Biotechnology and the other siRNA (siRNA2) (sense, 5′-3′ GCACUUAGAUGGGAAACUUTT and antisense, 5′-3′ AAGUUUCCCAUCUAAGUGCTT) was purchased from GenePharma. ATII cells ($2.5 \times 10^4$) were seeded in 24-cell plates and were cultured for 20 h; then, 20 pmol of siRNA was transfected with GenMute™ siRNA Transfection Reagent (SignaGen) according to the manufacturer's instructions. The sequences of non-targeting siRNAs were as follows: sense 5′-3′ UUCUCCGAACGUGUCACGUTT; antisense 5′-3′ ACGUGACACGUUCGGAGAATT. After 36 h of incubation, the cells were subjected to qRT-PCR or spore or latex bead internalization assays.

**Quantitative real-time PCR (qRT-PCR)**. RNA was isolated from each cell by using the CellAmp™ Direct RNA Prep Kit for RT-PCR (TaKaRa). First-strand cDNA was synthesized using PrimeScript™ RT Master Mix (TaKaRa). The reaction was performed in a 20 μl reaction mixture. cDNAs were stored at −20 °C. For qRT-PCR, the reaction mixture was prepared with TB Green® Premix Ex Taq™ II (Tli RNaseH Plus) (TaKaRa) according to the manufacturer's instructions and PCR was performed with a Prism 7000 Sequence Detection System (Applied Biosystems). The primers used for PCR are listed in Supplementary Table 1. PCR was performed under the following conditions: 95 °C for 30 s, 40 cycles of 95 °C for 5 s, and 60 °C for 30 s followed by a standard dissociation run to obtain melt curve profiles of the amplicons. Using the 2-ddCt method, relative internal mRNA expression of target genes was normalized to *GAPDH*.

**Preparation of heat-killed spores**. About $5 \times 10^8$ spores suspended in water were incubated at 62 °C for 60 min. To assay the viability of spores, $3 \times 10^7$ of spores were spotted on the YPAD plate and incubated at 30 °C for 2 days.

**Expression and purification of RAGE$^{1-341}$-His-FLAG**. For the expression and purification of RAGE$^{1-341}$-His-FLAG, pME-Hyg-RAGE$^{1-341}$-His-FLAG was transfected into HEK293 cells cultured in 15-cm dishes. After 12 h of incubation, the medium was exchanged and the cells were further cultured for 36 h. The medium was collected and centrifugation at 3000×g for 3 min to remove cells and debris. Then, the medium was passed through a high-performance His trap column (GE Healthcare). The column was washed with PBS and RAGE$^{1-341}$-His-FLAG was eluted by elution buffer (200 mM imidazole). For the pulldown of RAGE$^{1-341}$-His-FLAG with tRNA-bound beads, the truncated RAGE was further purified with anti-FLAG M2 affinity agarose beads (Sigma-Aldrich). 1 ml of the RAGE$^{1-341}$-His-FLAG eluted form high-performance His trap column was mixed with 50 μl of anti-FLAG M2 affinity agarose beads (Sigma-Aldrich) prewashed with PBS and rotated at 4 °C for 2 h. The agarose beads were washed with PBS four times and the recombinant RAGE$^{1-341}$-His-FLAG was eluted by PBS containing 500 μg/ml FLAG peptide.

**Analysis of the internalization of spores or latex beads by mammalian cells**. A round glass coverslip (14 mm in diameter) was placed on the bottom of 24-well plates. Cells ($1.5 \times 10^5$/well) were seeded in the plate and allowed to grow to a density $2-3 \times 10^5$ cells /well. The assay was performed in 500 μl of culture media supplemented with 0.025 μl of LysoTracker (Beyotime). Mammalian cells and purified spores or latex beads (2 μm in diameter, Sigma-Aldrich) were incubated in a $CO_2$ incubator at 37 °C for 1 h. Unless otherwise noted, spores or latex beads were incubated with cultured cells at $1.2 \times 10^7$ spores or beads/$4-6 \times 10^5$ cultured cells/ml in DMEM supplemented with LysoTracker. Then, the cells were placed on ice and washed twice with PBS, and intracellular spores or beads were analyzed under a fluorescence microscope. For RFP or RFP fusion expressing cells (Fig. 5a, b and Supplementary Fig. 7b), at least 50 cultured cells were analyzed. For the other assay, at least 100 cultured cells were analyzed. Spores observed in LysoTracker-positive compartments were defined as internalized particles. For the competitive inhibition assay with polynucleotides, RNA or DNA was added to the assay media at a

concentration of 90 µg/ml prior to the addition of spores. Wortmannin, picea-tannol, or latrunculin A were added at the final concentration of 100 nM, 50 µM, or 10 nM, respectively, 30 min before the addition of spores or beads.

For the analysis of the size of internalized spores, NIS-Element AR software (Nikon) was used to measure the longest dimension of spores. Forty spores were analyzed.

**Flow cytometry.** Cells ($2 \times 10^5$/well) were seeded in 12-well plates and allowed to grow to a density of $4$–$6 \times 10^5$ cells /well. The cells were incubated with $1.2 \times 10^7$ spores expressing GFP in a $CO_2$ incubator at 37 °C for 1 h. Then, the cells were treated with trypsin for 5 min and washed twice with PBS. The samples were analyzed by BD FACS AriaIII (BD). The data were analyzed using FlowJo X (BD).

**Induction of apoptosis and analysis of the internalization of apoptotic cells by mammalian cells.** Jurkat cells were harvested from culture and resuspended at a density of $10^6$/ml in RPMI 1640 containing 10% FCS. Apoptosis was induced for 12 h with 1.0 µM staurosporine (Beyotime). After 12 h of incubation, apoptotic cells were centrifuged at $860 \times g$ for 3 min at 4 °C and washed twice with PBS.

A round glass coverslip (14 mm in diameter) was placed on the bottom of 24-well plates. HEK293T or HEK293T *RAGE* KO cells ($1.5 \times 10^5$/well) were seeded in the plate containing 500 µl of DMEM and allowed to grow to a density of $2$–$3 \times 10^5$ cells/well. Apoptotic Jurkat cells were added to the culture at $10^6$ apoptotic cells/ $4$–$6 \times 10^5$ wildtype or *RAGE* KO cells/ml. After 2 h of incubation in a $CO_2$ incubator at 37 °C, 0.025 µl of LysoTracker Red was added to the culture media and incubated for 10 min. Then, the cells were placed on ice and washed twice with PBS, and intracellular apoptotic cells were analyzed under a fluorescence microscope. Apoptotic cells with green fluorescence observed in LysoTracker-positive compartments in HEK293T or HEK293T *RAGE* KO cells were defined as internalized apoptotic cells. For the assay, at least 300 cultured cells were analyzed. NIS-Element AR software (Nikon) was used to analyze the size (the longest dimension of internalized cells) of apoptotic cells in the lysosome.

**Extraction of RNA from the spore wall.** A total of $2 \times 10^8$ spores suspended in 500 µl of 0.6 M NaCl were vortexed at room temperature for 30 s. After centrifugation at $21,500 \times g$ for 10 min at 4 °C, RNA was purified from the supernatant with RNAiso plus (TaKaRa) according to the manufacturer's instructions. The RNA concentration was measured by using a NanoDrop 2000/2000c (Thermo Fisher Scientific).

**Spore wall RNA sequencing and data processing.** Spore wall-bound RNA was prepared as described above. RNA sequencing was performed by Genesky Bio-technologies. The RNA sample was applied to Agilent 2100 Bioanalyzer (Agilent Technologies) to analyze its size distribution (Supplementary Fig. 10a). The RNA sample (without fragmentation) were ligated with Illumina adapters (120 nt) to form target ligation products and cDNAs were generated by reverse transcription. Then, the cDNA fragments were amplified with PCR. The PCR products were subjected to 6% polyacrylamide gel electrophoresis to separate target DNA fragments and adapters (Supplementary Fig. 10b). Based on the Bioanalyzer result, smear DNA signal detected from 140 to 250 bp in the polyacrylamide gel was cut out and DNA fragments were recovered from the gel. The DNA fragments were sequenced on an Illumina MiSeq Benchtop Sequencer (Illumina). The sequence data have been deposited in the NCBI Sequence Read Archive database with accession number PRJNA748575. The distribution of RNA length was analyzed with FastQC[58].

Paired-end MiSeq reads were adapter trimmed (Trimmomatic[59]) and overlapping paired-end reads merged using PEAR[60]. Reads were mapped to the genome using BWA-MEM[61] and the fraction of reads mapping to each class of gene was determined using the GFF file (column 3) from the R64 genome release from SGD[62].

**Preparation of polynucleotide-bound salt-washed spores and polynucleotides-and/or proteins-bound beads.** For the preparation of polynucleotides-bound salt-washed spores, spores were washed one time with 0.6 M NaCl and twice with water. A total of $2 \times 10^8$ spores suspended in 1 ml water were incubated with polynucleotides at 4 °C for 24 h with rotation. Then, the spores were centrifuged at $1000 \times g$ for 5 min at 4 °C and washed twice with water. The number of spores were counted with a platelet counter.

For the preparation of DNA-bound latex beads, 40 µl of latex beads were washed twice with water. A total of $2 \times 10^8$ beads suspended in water were incubated with spore wall RNA, tRNA (Sigma-Aldrich), or DNA1 with rotation at 4 °C for 24 h. The number of beads were counted with a platelet counter. Then, the beads were centrifuged at $21,500 \times g$ for 5 min at 4 °C and washed twice with water. For the preparation of DNA beads (DNA1-bound beads), $2 \times 10^8$ latex beads were incubated with 200 ng of DNA1 in 1 ml water at 4 °C for 24 h and washed as described above.

To measure the amounts of polynucleotides bound to spores or beads, polynucleotide-bound spores or beads were suspended in 200 µl of 0.6 M NaCl and vortexed at room temperature for 30 s. Then, the spores or beads were centrifuged at $21,500 \times g$ for 5 min at 4 °C. RNA was purified from the supernatant with RNAiso

plus. DNA was purified by isopropanol precipitation. The RNA or DNA concentration was measured by using a NanoDrop 2000/2000c.

To generate DNA/HMGB1 or DNA/histone beads, $10^8$ DNA-bound beads suspended in 55 µl water were incubated with HMGB1 (Sangon Biotech) or histone (Sangon Biotech) with rotation at 4 °C for 24 h. The beads were centrifuged at $21,500 \times g$ for 5 min at 4 °C and washed twice with water. The amount of protein bound to beads was determined by subtracting the residual amount of the protein from that of the original amount. BCA kit (Beyotime) was used to measure protein amounts. To generate HMGB1-, histone-, or BSA-bound latex beads, first, $2 \times 10^8$ latex beads were incubated in 500 µl of water supplemented with 4% glutaraldehyde (Aladdin) at 30 °C for 2 h and washed twice with water. A total of $10^8$ glutaraldehyde-modified beads were incubated with HMGB1, histone, or BSA (Sangon Biotech) in 55 µl of water at 4 °C for 24 h with rotation. Then, the beads were washed twice with water.

**Pulldown of RAGE$^{1-341}$-His-FLAG with tRNA-bound beads.** For the preparation of tRNA-bound beads, $2 \times 10^8$ latex beads were incubated with 200 ng tRNA in 1 ml water at 4 °C for 24 h and washed twice with water. Seven hundred micrograms of purified RAGE$^{1-341}$-His-FLAG (after purification with anti-FLAG M2 affinity agarose beads) was incubated with $1.5 \times 10^8$ bare beads or tRNA-bound beads in 50 µl PBS at 4 °C for 2 h with rotation. Then, the beads were washed three times with PBS. The protein attached to the beads was eluted by boiling in 50 µl of SDS sample buffer and 10 µl of the samples were analyzed by western blotting.

**Assay for RNA binding to RAGE$^{1-341}$-His -FLAG.** One milliliter of the RAGE$^{1-341}$-His-FLAG eluted from high-performance His trap column was incubated with 50 µl of prewashed anti-FLAG M2 affinity agarose beads (Sigma-Aldrich) to attach the truncated RAGE to the agarose beads. The beads were washed three times with PBS and suspended in 50 µl RNase-free water. Then, 20 µl of RAGE$^{1-341}$-His-FLAG-bound agarose beads was mixed with 1.5 µg RNA fused to cyanine 3 (Cy3) at the 5′-end incubated in 60 µl of RNase-free water. The mixture was incubated at 4 °C for 2 h. After washing three times with RNase-free water, the beads were suspended in 200 µl RNase-free water, and 150 µl of the sample was applied to Synergy H4 Hybrid multi-mode microplate reader with 570 nm excitation and 650 nm emission for fluorescence quantification.

**Pulldown of RAGE with histone-bound beads.** A total of $6 \times 10^8$ latex beads were washed twice with water and incubated with 4% glutaraldehyde at 30 °C for 2 h. After centrifugation at $21,500 \times g$ for 5 min at 4 °C, the beads were washed twice with water. Glutaraldehyde-modified beads ($3 \times 10^8$) were incubated with 80 µg of histone or BSA at 4 °C in 100 µl of water for 12 h with rotation. After washing twice with water, the beads were incubated with 10% BSA at 4 °C for 1 h to block free glutaraldehyde.

HEK293T cells cultured in 6 cm dishes were transfected with pME-mEGFP or pME-RAGE-mEGFP using Lipofectamine 2000 and incubated for 36 h. Then, the cells were washed with PBS and incubated with 500 µl of lysis buffer (25 mM HEPES, pH 7.4, 150 mM NaCl, 0.1% NP40) supplemented with 5 µl of EDTA-Free protease inhibitor cocktails (MCE) on ice for 30 min. After centrifugation at $21,500 \times g$ for 10 min at 4 °C, 100 µl of the supernatant was incubated with $1.5 \times 10^8$ histone- or BSA-bound beads with rotation. Then, the beads were washed four times with lysis buffer. The proteins attached to the beads were eluted by boiling in 50 µl of SDS sample buffer (50 mM Tris-HCl pH 6.8, 2% SDS, 0.1% Bromophenol blue, 10% glycerol, and 1% 1-mercapto-11-hydroxy-3,6,9-trioxaundecane) and 10 µl of the samples were analyzed by western blotting. One percent of the lysate after centrifugation was used as the input sample for western blotting.

**Western blotting analysis.** To detect endogenous RAGE, HEK293T and HEK293T *RAGE* KO cells cultured in 10 cm dishes were suspended in 500 µl of homogenization buffer (10 mM HEPES, pH 7.4, 0.22 M mannitol, and 0.07 M sucrose) supplemented with 5 µl of EDTA-Free protease inhibitor cocktails (MCE). Cells were homogenized on ice 40 times. The homogenates were transferred to 1.5 ml tubes and centrifuged at $1000 \times g$ for 5 min at 4 °C. The supernatants were further centrifuged at $100,000 \times g$ for 1 h at 4 °C. The pellets were resuspended in a 100 µl homogenization buffer. Then, 10 µl of samples mixed with SDS sample buffer were boiled and subjected to 10% SDS-PAGE. To detect a loading control, GLUT1, 10 µl of the samples without boiling were subjected to 10% SDS-PAGE.

To detect the fusion of tagRFP to RAGE, TLR3, TLR7, TLR8, or TLR9, HEK293 cells harboring the pME-TLR3-tagRFP, pME-TLR7-tagRFP, pME-TLR8-tagRFP, or pME-TLR9-tagRFP plasmids were suspended in 500 µl of lysis buffer supplemented with protease inhibitor cocktails on ice for 30 min. After centrifugation at $21,500 \times g$ for 10 min at 4 °C, 20 µg of these samples was suspended in an SDS sample buffer. After the samples were boiled, they were subjected to 6% SDS-PAGE. After SDS-PAGE, the proteins were transferred to PVDF membranes (Bio-Rad). The membranes were blocked in 5% milk (Sangon Biotech) in TBST buffer (10 mM Tris-HCl, pH 7.5, 150 mM NaCl, and 0.05% (v/v) Tween-20) and probed with the appropriate antibodies diluted in QuickBlock™ Primary Antibody Dilution Buffer (Beyotime) for 1 h at room temperature or overnight at 4 °C. After washing with TBST three times, the membranes were incubated with HRP-conjugated secondary antibodies diluted in 5% milk in TBST buffer for 1 h at

room temperature and washed three times in TBST buffer. The following primary antibodies were used: rabbit anti-RAGE (Abcam, 1: 2000); rabbit anti-SYK (Abcam, 1: 3000); rabbit anti-RFP (InvivoGen, 1:3000); mouse anti-GFP (TransGen Biotech, 1:3000); rabbit anti-GLUT1 (Abcam, 1:4000); mouse anti-actin (TransGen Biotech, 1:3000); mouse anti-His (TransGen Biotech, 1:5000). Primary antibodies were detected using the secondary anti-mouse IgG HRP-linked (TransGen Biotech, 1:5000), or anti-rabbit IgG HRP-linked (TransGen Biotech, 1:5000). Signals were detected with ECL Substrate (Bio-Rad). Images were captured using a Tanon 5200 Automatic Chemiluminescence Image Analysis System.

**Microscopy**. Microscopy images were obtained using a Nikon C2 Eclipse Ti-E inverted microscope with a DS-Ri camera (live imaging and yeast images) or a Nikon C2 Eclipse Ti-E inverted confocal laser scanning microscope (the other images) equipped with NIS-Element AR software.

For live imaging, cells transiently transfected with RAGE-mEGFP were cultured on glass-bottom dishes with 1.5 ml of DMEM. When the cell density reached ~80% confluence, 0.06 μl of LysoTracker Red was added to the culture media and incubated for 20 min. Then, the culture media was exchanged with prewarmed DMEM-supplemented spores $(1.5 \times 10^7/\text{ml})$. After 20 min incubation, the dish was subjected to live imaging.

Quantitative analysis of RAGE-RFP in phagocytic cup and plasma membrane was performed as followings. HEK293T cells expressing RAGE-RFP were incubated with spores $(1.2 \times 10^7 \text{ spores}/4\text{–}6 \times 10^5 \text{ cultured cells/ml})$ for 40 min. After washing with PBS twice, the cells were fixed with 4% (vol/vol) paraformaldehyde for 20 min at room temperature. The fixed cells were washed with PBS twice. Fluorescent intensities were measured in certain selected areas $(32 \times 55 \text{ pixels})$ in the phagocytic cup (plasma membrane engulfing spores) and those adjacent to each phagocytic cup. The mean pixel intensities of the selected areas were obtained with NIS-Element AR software.

**Statistics and reproducibility**. Statistical analysis of the data was performed using GraphPad Prism 8 software (GraphPad Software). All values are presented as the means ± SEM ($n$ as indicated in the figure legends). Statistical significance was determined with a two-tailed unpaired Student's $t$-test. $P < 0.05$ was considered statistically significant.

**Reporting summary**. Further information on research design is available in the Nature Research Reporting Summary linked to this article.

## Data availability
The data supporting the findings of this study are available within the Article and its Supplementary Information. The RNA-sequencing data have been deposited in the NCBI Sequence Read Archive database (accession number PRJNA748575). Source data of the graphs presented in the figures are available in Supplementary Data 1. Unedited immunoblots and gels are provided as Supplementary Figs. 11–13 in the Supplementary Information file. Further relevant data were available from corresponding authors upon reasonable request.

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

## Acknowledgements

We are grateful to Aaron Neiman (Stony Brook University) for his comments on the manuscript. This work was supported by the National first-class discipline program of Light Industry Technology and Engineering (LITE2018-015), Project 111 (111-2-06), Collaborative Innovation Center of Jiangsu Modern Industrial Fermentation, International Joint Research Laboratory for the production of therapeutic glycoproteins at Jiangnan University to X.-D.G., Fundamental Research Funds for the Central Universities to H.N. (JUSRP51629B), and NSFC grants to H.N. (32071467) and X.-D.G. (21778023).

## Author contributions

Conceptualization, H.N., H.T., and Y.Y.; methodology, Y.Y., H.N., and M.F.; validation, Y.Y., G.L., and H.N.; formal analysis, Y.Y., L.B.C., and H.N.; investigation, Y.Y., H.N., G.L., F.L., C.S., and K.L.; writing—original draft preparation, Y.Y. and H.N.; writing—review and editing, H.N., Y.Y., H.T., M.F, F.L., and X.-D.G.; supervision and funding acquisition X.-D.G. and H.N.; All authors have read and agreed to the published version of the manuscript.

## Competing interests

The authors declare no competing interests.
