## [Peer Review File · Communications Biology]

Reviewers' comments:

Reviewer #1 (Remarks to the Author):

In this manuscript the authors aimed to investigate the phagocytosis of *S. cerevisiae* spores in nonprofessional phagocytes (NPPs). The main finding is that this particular process in NPPs is mediated by RAGE receptor recognizing RNA fragments attached to the spores of *S. cerevisiae* that results in efficient internalization of spores by NPPs. The ability of RAGE receptor to bind RNA is not a new phenomenon and has already been described in several scientific publications (e.g. doi: 10.4049/jimmunol.1502169). However, the binding of RNA attached to the spore wall of *S. cerevisiae* to RAGE receptor as a mechanism underlying the phagocytosis by NPPs is rather novel mechanism. The manuscript is well written, the text is clear to read and understand. Experimental data is clearly presented, analyzed and in many cases also interpreted correctly. However, in my opinion some important controls and experimental data are missing. Here are my comments:

1. Line 84-86 – the authors conclude that internalization of spores was not mediated by antibodies and complements, since HEK293T cells internalized spores also in the absence of serum. This statement is only half-true and leads to wrong conclusions. FCS, if the same used as for the cell culture (and this is not mentioned in Materials and Methods), should be heat-inactivated and thereby containing inactivated components of the complement system. To check, whether the engulfment of spores is mediated by immunoglobulins and complements additional opsonisation step with non heat-inactivated human (for HEK) or murine (ATII cells) serum should be included prior to exposition of spores to HEK cells. This needs to be corrected/rephrase.

2. Figure 3d and 3e - there are some differences between binding of spore RNA and tRNA or DNA1 to latex beads (Fig. 3d), but it doesn't correlate with the differences in number of beads internalized per HEK cell. How can this be explained? Here is also important control missing – the number of untreated beads which are internalized per HEK cell. Please include that, otherwise it is hard to interpret this results correctly and conclude that polynucleotide coating of beads enhances its internalization by HEK cells. Such a control is first included in Fig. 4c .

3. Fig. 4a – I miss here the values (number of spores internalized) for the initial HEK293 cells (prior to transfection). This is mostly because the values vary a lot - in Fig. 1c or 1d it's about 1, in Fig. 1e even 1.5, but in Fig. 4a it is only 0.5. How many cells phagocytized the spores? The proportion of HEK cells phagocytizing the spores is only presented in the Fig. 1b and for the conditions used in all experiments it's about 70%. What was the reason for choosing the lowest concentration and not one of those leading to 85-100% of cells with internalized spores? And then again - does it mean that the transfection with RFP first decreased the phagocytosis rate and subsequent overexpression of RAGE-RFP restored it? But in Fig. 4b it is again about 1.

4. Did the authors analyzed the expression of TLR3, TLR7, TLR8 and (the most important) RAGE in RFP cells? This (RAGE expression in RFP and in RAGE-RFP cells) is actually missing in the Suppl. Fig.2b. The values in Fig. 4c differ also dramatically (range 0.025-0.2). What is the reason for such a huge variation? Since the authors use only microscope analysis to determine the phagocytosis rate, it would be necessary to confirm at least some of the findings (e.g. results shown in Fig.2a or 2i and 4b) findings using an alternative method allowing the analysis of phagocytosis in a larger number of cells (e.g. ImageStream or FACS).

Reviewer #2 (Remarks to the Author):

In this study, the authors found that yeast spores can be efficiently internalized by HEK293 cells and the RNA fragments attached to the spore wall serve as ligands to induce spore internalization. AS an intrinsic mechanism by which NPPs internalize macromolecules, the phagocytosis of spores was found to be mediated by RAGE. However, there are several questions or shortcomings need to be addressed.

Major:

1. As microparticles in vitro, yeast spores were used to test the internalization in epithelial cells. What's the biological significance for this phenomenon? I suggest the authors to explain further in the discussion.

2. In this manuscript, especially in the results of Fig. 4, the authors conclude that "spores are internalized via RAGE-mediated phagocytosis" and "RNA fragments attached to the spore wall

serve as ligands to induce spore internalization". However, there lacks direct evidence for the association of RAGE and RNA.

3. As mentioned in line 76-79 (Fig1d, e), the authors indicate the significance of dityrosine on spores in the internalization of spores. However, what is the relation between dityrosine and RNA which is proved to be a main ligand to mediate internalization of spores?

4. Generally, receptor-mediated internalization is saturable. In the results shown in Fig1b, the internalization of spores in HEK293 cells is far from saturation. The authors should provide more groups to see the saturate concentration of spore internalization.

5. According to Fig1cd, it's hard to draw the conclusion of "Vegetative cells and dit1Δ spores were poorly internalized in HEK293T cells" (line 64-65).

Minor:

1. The WB result in supplementary-Fig2b is unclear.

2. The concentration of spores in Fig1a should be provided.

3. Line170-171: that conclusion cannot be got from different experiments.

4. RAGE KO HEK293T cells are the key model in this study. Please cite more literatures such as Cell Death Differ 2014; 21(8): 1229-1239. to show the importance of RAGE in internalization.

Reviewer #3 (Remarks to the Author):

The Receptor for Advanced Glycation End-Products (RAGE), is a multiligand receptor that mediates DAMPs signalling and functions as a scavenger receptor. It has been associated with inflammatory and anti-inflammatory responses and connected to numerous diseases. RAGE is modestly expressed in several cell types, but at very high levels in the lungs, particularly in the alveolar type I epithelial cells. In macrophages RAGE enhances the phagocytosis of pathogens and apoptotic cells. However, the mechanism by which RAGE modulates phagocytosis is unknown.

In this manuscript the authors report RAGE mediates phagocytosis in non-professional phagocytes, Type II alveolar epithelial cells and HEK cells. They show the internalization of *S. cerevisiae* ascospores and latex beads, which were naturally or artificially covered with RAGE ligands, as well as apoptotic cells. The authors used *S. cerevisiae* spores for their studies and report the intracellular germination of spores, which seems disconnected from the rest of the manuscript. It is unclear why they chose this target, or validate their findings, over spores from a pathogenic organism. Through a series of enzymatic treatments and using dit1Δ cell wall mutants, the authors found that RNA from the ascus adsorbed to the spores is recognized by RAGE. The authors propose that the cell wall's dityrosine layer and cell wall proteins actively bind RNA to spores.

Although I believe this speculation deviates from the main focus of the study, the authors should consult doi: 10.1074/jbc.m117.786202, where it is shown that proteinase K treatments destabilize the assembly of the dityrosine-enriched coating over the chitosan layer.

It is difficult to understand why did the authors decided to carry out the RNA sequencing studies. How is this contributing to the understanding of non-professional phagocytosis?

The authors show that salt-washed and RNase treated spores are no longer internalized by cells, unless the RNA is restored or replaced by other forms of RNA or DNA, albeit yielding lower levels of uptake efficiency. Similar experiments showed that epithelial cells can internalize polynucleotide-covered latex beads. To prove the involvement of RAGE in the phagocytosis of these particles the authors resorted to over-expression and knock-down of RAGE as well as to the use of pharmacological inhibitors of actin polymerization. Other experiments assessed the internalization of particles covered with known RAGE ligands.

In this way, the authors collected correlational evidence in support of RAGE mediated phagocytosis in non-professional phagocytes. Although the topic of this manuscript is relevant, the evidence presented is too preliminary and the experimental design seems unfocused. The study lacks mechanistic insight, as it does not attempt to explain how RAGE may contribute to phagocytosis in non-professional markers.

Response to referees

I would like to thank all the reviewers for reviewing our manuscript and valuable comments. Response to reviewers are follows.

Reviewer #1:

1. Line 84-86 – the authors conclude that internalization of spores was not mediated by antibodies and complements, since HEK293T cells internalized spores also in the absence of serum. This statement is only half-true and leads to wrong conclusions. FCS, if the same used as for the cell culture (and this is not mentioned in Materials and Methods), should be heat-inactivated and thereby containing inactivated components of the complement system. To check, whether the engulfment of spores is mediated by immunoglobulins and complements additional opsonisation step with non heat-inactivated human (for HEK) or murine (A2II cells) serum should be included prior to exposition of spores to HEK cells. This needs to be corrected/rephrase.

Response:

We understood that our result is not sufficient to conclude that internalization of spores is not mediated by antibodies and the complements. However, it is not the main object for this study to find whether antibodies and the complements are involved in the phagocytic process in NPPs. In addition, it would not be simple to prove that antibodies and the complements are involved in the process. Thus, we rephrase the manuscript as “internalization of spores occurs even in the absence of antibodies and the complement system” (line 87-90 in the track changes file; line 81-84 in the PDF file). In the previous manuscript, we did not mention that mammalian cells were cultured in media supplemented with heat-inactivated FCS. This information has been included in Materials and methods (line 343 in the track changes file; line 305 in the PDF file).

2. Figure 3d and 3e - there are some differences between binding of spore RNA and tRNA or DNA1 to latex beads (Fig. 3d), but it doesn't correlate with the differences in number of beads internalized per HEK cell. How can this be explained? Here is also important control missing – the number of untreated beads which are internalized per HEK cell. Please include that, otherwise it is hard to interpret this results correctly and conclude that polynucleotide coating of beads enhances its internalization by HEK cells. Such a control is first included in Fig. 4c .

Response:

Polynucleotides can induce internalization of latex beads. However, the levels of internalization of polynucleotide-bound latex beads were significantly lower than those of spores or polynucleotide-bound high-salt-washed spores. Presumably because of the low internalization efficiency, no significant differences were found between the internalization levels of spore wall-derived RNA-bound latex beads and tRNA- or DNA1-bound latex beads. This explanation has been included in the manuscript (line 141-143 in the track changes file; line 131-134 in the PDF file). In Fig. 3a and b (Fig 3d and e, respectively, in the original manuscript), the phagocytosis assay was performed with the constant number (1.2×10^7) of beads; in Fig. 3b, for example, amounts of polynucleotides attached to beads were changed and internalization was assayed. The causes for this confusion are our mistake in writing (figure legend) and inappropriate labeling of the X-axis. Thus, we have revised the figure legend (for Fig. 3b, line 764-765 in the track changes

file and line 705-706 in the PDF file) and labeling for all similar graphs (Fig. 1b, Fig. 3, Fig. 4a, b, Supplementary Fig. 6, Supplementary Fig. 7a, b, d, e).

3. Fig. 4a – I miss here the values (number of spores internalized) for the initial HEK293 cells (prior to transfection). This is mostly because the values vary a lot - in Fig. 1c or 1d it's about 1, in Fig. 1e even 1.5, but in Fig. 4a it is only 0.5. How many cells phagocytized the spores? The proportion of HEK cells phagocytizing the spores is only presented in the Fig. 1b and for the conditions used in all experiments it's about 70%. What was the reason for choosing the lowest concentration and not one of those leading to 85-100% of cells with internalized spores? And then again - does it mean that the transfection with RFP first decreased the phagocytosis rate and subsequent overexpression of RAGE-RFP restored it? But in Fig. 4b it is again about 1.

Response:

We used HEK293T cell in most of experiments. However, in Fig 4a, we used HEK293 cell (not HEK293“T”). HEK293 cells were used in this experiment because this cell line exhibited lower phagocytic efficiency than HEK293T; we speculated that phagocytic levels in HEK293 could be enhanced by overexpression of the receptor. Now, we have added this description in the text (line 188-190 in the track changes file; line 155-157 in the PDF file). We performed phagocytosis assay in an unsaturated condition. Therefore, the assays were performed with defined conditions. We think that phagocytosis efficiency can be evaluated with this condition.

4. Did the authors analyzed the expression of TLR3, TLR7, TLR8 and (the most important) RAGE in RFP cells? This (RAGE expression in RFP and in RAGE-RFP cells) is actually missing in the Suppl. Fig.2b. The values in Fig. 4c differ also dramatically (range 0.025-0.2). What is the reason for such a huge variation? Since the authors use only microscope analysis to determine the phagocytosis rate, it would be necessary to confirm at least some of the findings (e.g. results shown in Fig.2a or 2i and 4b) findings using an alternative method allowing the analysis of phagocytosis in a larger number of cells (e.g. ImageStream or FACS).

Response:

In Fig 4a (in the revised manuscript, the figure has been moved to Fig. 5a), plasmids were transfected transiently in HEK293 cells. Thus, endogenous RAGE levels should be same between RFP and RAGE-RFP expressing cells. Expression levels of transfected genes (RFP fusion genes) were analyzed by western blotting with anti-RFP antibody (Fig 4a). In addition, we performed qRT-PCR assay to analyze expression levels of RAGE mRNA in HEK293 and HEK293T PCR (as described above, the experiment shown in Fig. 4a was performed with HEK293). The result shows that RAGE expression in HEK293T is higher than HEK293, which is consistent with the result that HEK293T can phagocytose more than HEK293. This result has been shown in Supplementary Fig. 5a and described in the text (line 189-193 in the track changes file; line 156-160 in the PDF file).

The value in Fig. 4c (the figure has been moved to Fig. 5c) is significantly low because the experiment was performed with latex beads (not spores). The values are consistent with other results (for example Fig. 3b).

As the reviewer suggested, we performed FACS analysis for Fig. 2a and phagocytosis of spores by HEK293T *RAGE* KO cells (the reviewer also mentioned about Fig 2i but this is a study for polynucleotide binding, thus FACS analysis was not performed for this study). The results are consistent with the microscopic analysis, which corroborate our results. The FACS results have been shown in Supplementary Fig. 2 and 5d and described in line 103-106 in the track changes file (line 96-100 in the PDF file) and line 196-197 in the track changes file (line 163-164 in the PDF file).

Reviewer #2:

1. As microparticles in vitro, yeast spores were used to test the internalization in epithelial cells. What's the biological significance for this phenomenon? I suggest the authors to explain further in the discussion.

Response:

We have modified discussion to emphasize possible role of RAGE in vivo (line 322-325 in the track changes file; line 285-288 in the PDF file).

2. In this manuscript, especially in the results of Fig. 4, the authors conclude that “spores are internalized via RAGE-mediated phagocytosis” and “RNA fragments attached to the spore wall serve as ligands to induce spore internalization”. However, there lacks direct evidence for the association of RAGE and RNA.

Response:

We performed two in vitro binding assays to show direct binding between RNA and RAGE. The results have been shown in Fig. 5d, e and described in the text (line 206-215 in the track changes file; line 173-181 in the PDF file).

3. As mentioned in line 76-79 (Fig1d, e), the authors indicate the significance of dityrosine on spores in the internalization of spores. However, what is the relation between dityrosine and RNA which is proved to be a main ligand to mediate internalization of spores?

Response:

The dityrosine layer is the outermost layer of the spore wall. This layer is required for binding of RNA. To avoid misunderstanding, we have modified the manuscript (line 95 in the track changes file; line 89 in the PDF file).

4. Generally, receptor-mediated internalization is saturable. In the results shown in Fig1b, the internalization of spores in HEK293 cells is far from saturation. The authors should provide more groups to see the saturate concentration of spore internalization.

Response:

According to the reviewer's suggestion, we have performed the experiment and revised Fig 1b. Now the result shows that phagocytosis by HEK293T cells can be saturated.

5. According to Fig1cd, it's hard to draw the conclusion of "Vegetative cells and *dit1*Δ spores were poorly internalized in HEK293T cells" (line 64-65).

Response:

Based on the reviewer's suggestion, the manuscript has been revised as "Compared to wild-type spores, the internalization levels of vegetative cells and *dit1Δ* spores by HEK293T cells were significantly lower" (line 67-68 in the track changes file; line 65-66 in the PDF file).

Minor:

1. The WB result in supplementary-Fig2b is unclear.

Response:

We have revised the figure. The figure has been moved to Supplementary Fig. 5c.

2. The concentration of spores in Fig1a should be provided.

Response:

We have described the concentration of spores in the legend of Fig 1a (line 720-721 in the track changes file; line 679-680 in the PDF file).

3. Line170-171: that conclusion cannot be got from different experiments.

Response:

We have revised the text as "Compared to wild-type HEK293T cells, the internalization levels of HMGB1 beads were significantly decreased in *RAGE* KO cells (Fig 7b). These results suggest that HMGB1 induces phagocytosis by NPPs". (line 236-237 in the track changes file; line 202-203 in the PDF file).

4. RAGE KO HEK293T cells are the key model in this study. Please cite more literatures such as Cell Death Differ 2014; 21(8): 1229-1239. to show the importance of RAGE in internalization.

Response:

Based on the reviewer's suggestion, we have revised the text (line 47-48 in the track changes file; line 46-47 in the PDF file).

Reviewer #3:

It is unclear why they chose this target, or validate their findings, over spores from a pathogenic organism.

Response:

We found that spores are internalized in NPPs during the course of our study to use spores of the yeast *Saccharomyces cerevisiae* as microparticles as described in Introduction (line 41-43 in the track changes file; line 40-42 in the PDF file). Since *S. cerevisiae* is not a pathogenic organism, physiological significance of the spore's toxicity is unclear. Nevertheless, we interested in the phenomenon that spores are internalization in cultured cells and the basis of the internalization mechanism was further analyzed. As a result, we found the RAGE dependent

phagocytosis. In this study we report the phagocytic process. To avoid confusion, we have modified the manuscript (line 73-75 in the track changes file; line 71-73 in the PDF file).

Through a series of enzymatic treatments and using *dit1*Δ cell wall mutants, the authors found that RNA from the ascus adsorbed to the spores is recognized by RAGE. The authors propose that the cell wall's dityrosine layer and cell wall proteins actively bind RNA to spores. Although I believe this speculation deviates from the main focus of the study, the authors should consult doi:10.1074/jbc.m117.786202 , where it is shown that proteinase K treatments destabilize the assembly of the dityrosine-enriched coating over the chitosan layer.

Response:

The previous study showed that protein is required for assembly of dityrosine molecules. However, once the dityrosine layer is formed in the spore wall, dityrosine molecules are covalently attached to the spore wall (please refer modified introduction, line 37-38 in the track changes file; line 37-38 in the PDF file). Proteinaceous RNA-binding machinery described in this study is most likely attached to the dityrosine layer; however, the machinery is not required for assembly of dityrosine molecules. Thus, we think that our results and arguments are not contradictory to the previous report.

It is difficult to understand why did the authors decided to carry out the RNA sequencing studies. How is this contributing to the understanding of non-professional phagocytosis?

Response:

We performed RNA sequencing analysis because the mechanism of phagocytosis in NPPs was unknown, and thus, a unique RNA, which could induce phagocytosis, might attached to the spore wall. To make this point clear, we have reorganized the text (line 107-160 in the track changes file; line 101-151 in the PDF file) and figures (previous Fig. 2d, e, f has been moved to Supplementary Fig. 3a, b, c; previous Fig. 2g has been moved to Supplementary Fig. 4a; previous Fig. 2h has been moved to Fig 4a; previous Fig. 2i has been modified and moved to Supplementary Fig. 4b; data of previous Fig. 2j has been included in Supplementary Fig. 4b; previous Fig. 3a has been moved to Fig. 4b; previous Fig. 3b, c has been moved to Fig. 4c, d, respectively; previous Fig. 3d, e has been modified (DNA1 has been removed to avoid confusion) and moved to Fig. 3a, b, respectively; Fig. 5, 6, 7 have been moved to Fig. 6, 7, 8, respectively)

In addition, the reviewer criticizes that the lack of mechanistic insight of the phagocytosis process. Although we could not provide detailed mechanistic studies, we report that RAGE serves as a receptor to induce phagocytosis in this study. We would like to present further mechanistic analyses in a separated study.

Reviewers' comments:

Reviewer #1 (Remarks to the Author):

All my concerns were addressed by the authors. I have no further comments.

Reviewer #2 (Remarks to the Author):

No further questions.

Reviewer #3 (Remarks to the Author):

Despite the modifications introduced to the original manuscript, the revised version still lacks enough mechanistic insights and originality of data. Indeed, most of the findings described in this work have been already published in previous studies. For example, the binding of RAGE to RNA in a sequence independent manner and its role increasing RNA endocytosis were reported in <https://doi.org/10.4049/jimmunol.1502169>. The role of RAGE enhancing efferocytosis in macrophages and HEK cells was published in DOI: <https://doi.org/10.4049/jimmunol.1004134>. The same authors reported that RAGE binds phosphatidylinositol to enhance the attachment of apoptotic cells to phagocytes. Furthermore, [doi:10.1038/embor.2011.28](https://doi.org/10.1038/embor.2011.28) reported that RAGE interacts with the actin nucleation forming mDia1 to enhance efferocytosis and speculated about a downstream Rac1 activation mediating this process.

Several important methodological problems remain in the revised manuscript:

-The experimental design utilized in the manuscript to assess phagocytosis doesn't differentiate attached from internalized particles. This may impact the reliability of the phagocytic efficiencies reported, considering the large numbers of spores per cell utilized in the assays, and the apparent low efficiency of internalization.

-The characterization of RAGE distribution during phagocytosis, a key piece of information to fulfill the objective of this study, is shown in Fig 5 and its associated video. Unfortunately, the imaging data provided are of very poor quality and instead of supporting the manuscript conclusions it shows no enrichment of RAGE in the apparent phagocytic cup or phagosome - This when RAGE levels in these structures are compared with those from unengaged plasma membrane. Furthermore, in their analysis the authors ignored the fact that all except for one, all the spores shown in the video are negative for RAGE.

- Beyond lysotracker accumulation COMMSBIO-21-3706A presents no additional data on cellular structures associated to a presumptive RAGE mediated non-professional phagocytosis (NPP) i.e., phagocytic cup and phagosome, hence providing a very limited characterization of the process.

-The pharmacological assays presented are simple not enough to decipher a possible RAGE-mediated phagocytic pathway and present technical limitations. For example, inhibiting PI3K with wortmannin blocks phagosomal maturation and hence acidification, this probably hampered the identification of phagosomes (wrongly referred as lysosome all along the manuscript) with lysotracker.

-The authors chose to inhibit Syk phosphorylation. Syk is expressed in myeloid cells, Is SYK also expressed in HEK cells? On the other hand the authors didn't explore the involvement of RAGE downstream signalling previously reported for endocytosis, cell migration and phagocytosis, as in the case of mDia 1 and Rho small GTPases. The authors explored the interaction with TLR receptors which are not phagocytic. Wouldn't be more appropriate to focus in Dectin-1 and scavenger receptors instead? If RAGE is responsible for NPP, as sustained by the authors, it is important to determine the level of expression of this receptor in primary NP.

In summary, in its current form COMMSBIO-21-3706A is neither advancing our understanding on RAGE function in NPP, nor our understanding of NPP. In my opinion this study is not contributing enough novel information to merit its publication in this journal.

Response to reviewers

I would like to thank all the reviewers for taking a time to review our revised manuscript and giving us valuable comments. Responses to reviewer #3's comments are follows.

1. The experimental design utilized in the manuscript to assess phagocytosis doesn't differentiate attached from internalized particles. This may impact in the reliability of the phagocytic efficiencies reported, considering the large numbers of spores per cell utilized in the assays, and the apparent low efficiency of internalization.

Response:

In the FACS assay, it is possible that spores attached to the surface of mammalian cells are counted as internalized ones. Therefore, we performed the microscopy assay using LysoTracker to find internalized spores/beads. These assays have exhibited similar results, which corroborate our proposition.

2. The characterization of RAGE distribution during phagocytosis, a key piece of information to fulfill the objective of this study, is shown in Fig 5 and its associated video. Unfortunately, the imaging data provided are of very poor quality and instead of supporting the manuscript conclusions it shows no enrichment of RAGE in the apparent phagocytic cup or phagosome - This when RAGE levels in these structures are compared with those from unengaged plasma membrane. Furthermore, in their analysis the authors ignored the fact that all except for one, all the spores shown in the video are negative for RAGE.

Response:

We have added data showing that RAGE is recruited to phagocytic cup when spores are internalized (Supplementary Fig 7f). The result has been described in the text (line 189-192 in the track changes file; line 187-190 in the PDF file).

While the referee pointed out that only one spore is engulfed by RAGE in Movie 1 and Fig 5f, the movie shows another spore enclosed by RAGE. Some internalized spores may lack RAGE presumably because RAGE is transported to other endocytic organelles or plasma membrane (or the phagosome may be out of focus). Thus, it is possible that not all internalized spores are enclosed by RAGE.

3. Beyond lysotracker accumulation COMMSBIO-21-3706A presents no additional data on cellular structures associated to a presumptive RAGE mediated non-professional phagocytosis (NPP) i.e., phagocytic cup and phagosome, hence providing a very limited characterization of the process.

Response:

We are interested in detailed mechanism of RAGE-mediated phagocytosis in NPPs. However, the primary point for this paper is to report that RAGE serves as a receptor to induce phagocytosis in NPPs. We would like to present further analysis of the phagocytic process in a separated study.

4. The pharmacological assays presented are simple not enough to decipher a possible RAGE-mediated phagocytic pathway and present technical limitations. For example, inhibiting PI3K with wortmannin blocks phagosomal maturation and hence acidification, this probably hampered the identification of phagosomes (wrongly referred as lysosome all along the manuscript) with lysotracker.

Response:

PI3K inhibitor is also known to inhibit to phagosome formation in some phagocytic pathways including the one mediated by Fc γ receptor. We performed the FACS-based phagocytosis assay and the result suggest that HEK293T cells treated with the PI3K inhibitor exhibited a defect in internalization of spores. Inhibitors for actin polymerization and Syk also inhibited internalization process. The data has been shown in Supplementary Fig 2 and described in the text (line 81-83 in the track changes file; line 81-83 in the PDF file). According to this change, line 100-103 in the track changes file or line 100 in the PDF file has been modified.

5. The authors chose to inhibit Syk phosphorylation. Syk is expressed in myeloid cells, Is SYK also expressed in HEK cells? On the other hand the authors didn't explore the involvement of RAGE downstream signalling previously reported for endocytosis, cell migration and phagocytosis, as in the case of mDia 1 and Rho small GTPases. The authors explored the interaction with TLR receptors which are not phagocytic. Wouldn't be more appropriate to focus in Dectin-1 and scavenger receptors instead? If RAGE is responsible for NPP, as sustained by the authors, it is important to determine the level of expression of this receptor in primary NP.

Response:

To verify that SYK is involved in phagocytosis in HEK293T cells, we have detected SYK by western blotting (Supplementary Fig. 3a). Furthermore, we have added data that levels of phagocytic uptake of spores in HEK293T cells is decreased by SYK knockout (Supplementary Fig. 3b). The result has been described in the text (line 83-85 in the track changes file; line 83-85 in the PDF file). We would like to analyze other signaling molecules involved in the RAGE-mediated phagocytosis in a separated study.

Other changes

1. Another author (Kaiping Ling) has been added. She performed experiments regarding *SYK* knockout.

2. Syk has been changed to SYK.

3. In Supplementary Figure 4, a hyphen has been added before 'treated'.

4. In Supplementary Figure 7d right panel, 'percent of HEK293T cells' on the Y label has been change to 'percent of wt or *RAGE* KO cells'. In the legend, 'HEK293T cells' has been changed to 'HEK293T or HEK293T *RAGE* KO cells'.

5. Materials and Methods has been modified (line 341-343, 377-381, 350, and 544-549 in the track

changes file; line 339-341, 375-379, 528, and 542-547 in the PDF file).

Reviewers comments:

Referee 3:

The authors haven't answered my serious concerns about the originality of this research, nor have they provided additional mechanistic information to improve the manuscript. Unfortunately, the technical limitations remain, and the explanations provided by the authors are not sufficient to improve the quality of the data. For example, LysoTracker staining is not sufficient to determine a phagocytic index. LysoTracker labels intracellular acidic compartments, but not every phagosome. LysoTracker may also label extracellular spores, as unspecific labelling of microbial cells is a common problem with this dye, provided enough time of contact. Furthermore, the authors haven't address this reviewer's concern on the incompatibility of utilizing a pan PI3K inhibitor (which not only affects the uptake of particles > 3µm, but also the acidification of phagosomes), in combination with LysoTracker to determine phagocytosis. The authors clarified the point about SYK expression in HEK293, but no role for its involvement was investigated.

The quality of the new imaging data provided in SF 7 is poor. Its level of resolution is not enough to identify cellular structures and prevents extracting conclusions objectively. Also, as in video 1, SF 7f shows no RAGE enrichment around the spores. The levels of fluorescence are visually identical to those from RAGE in unengaged membrane.

Response to the reviewer

I would like to thank the reviewer for taking a time to review our revised manuscript and giving us valuable comments. Responses to reviewer #3's comments are follows.

1. LysoTracker staining is not sufficient to determine a phagocytic index. LysoTracker labels intracellular acidic compartments, but not every phagosome. LysoTracker may also label extracellular spores, as unspecific labelling of microbial cells is a common problem with this dye, provided enough time of contact.

Response:

In our experimental condition, spores or latex beads are not stained by LysoTracker. The evidences have been shown in Supplementary Fig. 1g. Accordingly, the text has been modified (line 65-66 and 116-117 in the track changes file; line 65-66 and 116-117 in the PDF file).

2. Furthermore, the authors haven't address this reviewer's concern on the incompatibility of utilizing a pan PI3K inhibitor (which not only affects the uptake of particles > 3um, but also the acidification of phagosomes), in combination with LysoTracker to determine phagocytosis.

Response:

As the reviewer pointed out, spores in immature phagosomes could not be found by the microscopy-based phagocytosis assay. However, to avoid such misunderstanding, we performed the FACS-based phagocytosis assay (Supplementary Fig. 2). The result shows that PI3K inhibitor inhibits internalization of spores.

3. The authors clarified the point about SYK expression in HEK293, but no role for its involvement was investigated. The quality of the new imaging data provided in SF 7 is poor. Its level of resolution is not enough to identify cellular structures and prevents extracting conclusions objectively. Also, as in video 1, SF 7f shows no RAGE enrichment around the spores. The levels of fluorescence are visually identical to those from RAGE in unengaged membrane.

Response:

We modified the text (line 186-193 in the track changes file; line 186-192 in the PDF file) as followings.

“To assess the localization of RAGE during internalization of spores, we performed time-lapse microscopy. This analysis showed that the spore was engulfed by green fluorescent protein (GFP) fused to RAGE before its internalization (Fig. 5f and Supplementary Video 1). RAGE-GFP was internalized together with the spores, and later, the GFP fusion- and spore-containing compartments were stained with LysoTracker (Fig. 5f and Supplementary Video 1). Levels of fluorescent intensities of RFP in certain areas in phagocytic cups (plasma membrane engulfing spores) was higher than those of areas adjacent to the phagocytic cup in the plasma membrane in fixed cells (Supplementary Fig 7f and g). This result suggests that RAGE-RFP is recruited to phagocytic cups.”

High quality pictures or movies (Fig. 5f and Supplementary Fig. 7f) may not provide further information of RAGE behavior and mechanism of the phagocytic process. Thus, the figures have not been modified in the revised manuscript. Nevertheless, we have provided a result showing that RAGE-RFP levels are higher than those in the neighboring plasma membrane area (Supplementary Fig. 7f). Based on this result, we have described that RAGE-RFP is recruited to phagocytic cups.